# FMRP deficiency leads to multifactorial dysregulation of splicing and mislocalization of MBNL1 to the cytoplasm

**Suna Jung**[1][◎], **Sneha Shah**[1][◎], **Geongoo Han**[2], **Joel D. Richter**[1]*

**1** Program in Molecular Medicine, University of Massachusetts Chan Medical School, Worcester, Massachusetts, United States of America, **2** Molecular Microbiology and Immunology, Brown University, Providence, Rhode Island, United States of America

◎ These authors contributed equally to this work.
* joel.richter@umassmed.edu

**Data Availability Statement:** All sequence files are available from the GEO database (accession number GSE207145). Underlying numerical values for the figures can be found in S3 Data.

## Abstract

Fragile X syndrome (FXS) is a neurodevelopmental disorder that is often modeled in *Fmr1* knockout mice where the RNA-binding protein FMRP is absent. Here, we show that in *Fmr1*-deficient mice, RNA mis-splicing occurs in several brain regions and peripheral tissues. To assess molecular mechanisms of splicing mis-regulation, we employed N2A cells depleted of *Fmr1*. In the absence of FMRP, RNA-specific exon skipping events are linked to the splicing factors hnRNPF, PTBP1, and MBNL1. FMRP regulates the translation of *Mbnl1* mRNA as well as *Mbnl1* RNA auto-splicing. Elevated *Mbnl1* auto-splicing in FMRP-deficient cells results in the loss of a nuclear localization signal (NLS)-containing exon. This in turn alters the nucleus-to-cytoplasm ratio of MBNL1. This redistribution of MBNL1 isoforms in *Fmr1*-deficient cells could result in downstream splicing changes in other RNAs. Indeed, further investigation revealed that splicing disruptions resulting from *Fmr1* depletion could be rescued by overexpression of nuclear MBNL1. Altered *Mbnl1* auto-splicing also occurs in human FXS postmortem brain. These data suggest that FMRP-controlled translation and RNA processing may cascade into a general dys-regulation of splicing in *Fmr1*-deficient cells.

## Introduction

Fragile X syndrome (FXS) is a neuro-developmental disorder characterized by mild to severe intellectual disability, speech and developmental delays, social impairment, perseveration, aggression, anxiety, and other maladies. FXS lies on the autism spectrum and is the most common single gene cause of autism. FXS is caused by an expansion of 200 or more CGG triplets in the 5′ untranslated region (UTR) of *FMR1*, which in turn induces DNA methylation and gene silencing. Loss of the *FMR1* gene product FMRP results in the disruption of neuronal circuitry and synaptic efficacy, which produces an array of neuro-pathological conditions [1–3]. FMRP, an RNA-binding protein present in probably all cells is frequently studied in mouse hippocampus, where several studies show that it represses protein synthesis [4–7]. This observation, in conjunction with results showing that FMRP co-sediments with polysomes in

**Funding:** This work was supported by National Institutes of Health/National Institute of General Medical Sciences (grant numbers GM046779, GM135087, GM149216, https://www.nigms.nih. gov/, funded to JDR), National Institutes of Health/ National Institute of Neurologic Disorders and Stroke (grant number NS132935, https://www. ninds.nih.gov/, funded to JDR), National Institutes of Health/National Center for Advancing Translational Sciences (grant number UL1-TR001453, https://ncats.nih.gov, funded to Katherine Luzuriaga with a pilot award to JDR), FRAXA Research Foundation grant (no identifying number, https://www.fraxa.org, funded to JDR), and FRAXA Research Foundation grant postdoctoral fellowship (no identifying number, https://www.fraxa.org, funded to SS). The funders had no role in study design, data collection and analysis, decision to publish, or preparation of the manuscript.

**Competing interests:** The authors have declared that no competing interests exist.

**Abbreviations:** FXS, fragile X syndrome; GO, Gene Ontology; KO, knockout; MBNL1, muscleblind like splicing regulator 1; MFI, mean fluorescence intensity; NLS, nuclear localization signal; TPM, transcripts per million; WT, wild type.

sucrose gradients [8,9] and that in UV CLIPs (crosslink-immunoprecipitation) mostly to coding regions of mRNA [5,10–12] suggests that it inhibits translation by impeding ribosome translocation. Indeed, it is now clear that at least 1 activity of FMRP is to stall ribosomes [5,7,13–15]. How this occurs is unclear, but it could involve codon bias or optimality [16,17], impairment of ribosome function [18], or formation of translationally quiescent subcellular granules [13].

One group of FMRP target RNAs encodes chromatin modifying enzymes [5,19,20]. The synthesis of several of these enzymes is inhibited by FMRP; in its absence, excessive levels of these chromatin proteins alter the epigenetic landscape, which in turn impairs cognitive function [19]. A few mRNAs encoding epigenetic factors associate with FMRP-stalled ribosomes [15]. One of these, *Setd2*, encodes an enzyme that establishes the histone modification H3K36me3, which is most often located in gene bodies [21,22]. In *Fmr1*-deficient mouse brain, SETD2 protein levels are elevated, which in turn alter the distribution of H3K36me3 chromatin marks. H3K36me3 has been linked to alternative pre-mRNA splicing [23–25], and indeed there is some correlation between the genes with recast H3K36me3 and altered splicing in *Fmr1*-deficient mouse hippocampus [15]. The observation that *Fmr1* deficiency results in hundreds of mis-splicing events prompted us to investigate both the prevalence and mechanism of FMRP-regulated nuclear pre-RNA processing.

We find that mis-splicing, mostly exon skipping, is widespread in *Fmr1*-deficient mice and occurs in all brain regions and peripheral tissues examined. To determine how FMRP might regulate splicing, we depleted *Fmr1* from mouse N2A cells, which resulted in hundreds of mis-splicing events. We focused on specific exons in 3 RNAs that are aberrantly skipped or included in *Fmr1*-deficient cells and mapped surrounding splicing factor consensus-binding sites. Splicing factors MBNL1, PTBP1, and hnRNPF are responsible for altered splicing in *Fmr1*-deficient cells. FMRP regulates the translation of 2 factors, MBNL1 and hnRNPQ. Moreover, *Mbnl1* RNA itself undergoes alternative splicing, which is impaired in *Fmr1*-deficient cells. In the absence of FMRP, a nuclear localization signal (NLS)-containing exon is frequently skipped, which alters the nucleus-cytoplasm distribution of MBNL1. This change in subcellular localization of MBNL1 may affect splicing decisions on other mRNAs. Notably, AS events affected by FMRP and MBNL1 exhibit a robust correlation and the ectopic expression of an MBNL1 isoform containing an NLS rescues approximately one-fifth of the disrupted splicing in *Fmr1*-depleted cells. In addition, *MBNL1* splicing is altered in human FXS postmortem cortex, suggesting that it could modify the brain proteome and thereby contribute to intellectual impairment and FXS.

## Results

### RNA splicing mis-regulation in *Fmr1* KO brain

Gene expression and RNA splicing are mis-regulated in the *Fmr1*-deficient mouse hippocampus [15] and FXS patient-derived blood samples [26]. To determine whether this mis-regulation occurs in other brain regions and in peripheral tissues from mice, we sequenced RNA from (*n* = 3, 2- to 3-month-old) WT and *Fmr1* KO hippocampus, cerebellum, and cortex, as well as liver, muscle, and testis (Fig 1A). Volcano plots show that hundreds of RNAs are up- or down-regulated in *Fmr1* KO cortex although fewer RNAs were similarly mis-regulated in hippocampus and cerebellum (log$_2$FC > 0.2 or < −0.2, FDR < 0.05, *n* = 3) (Fig 1B). A Venn diagram shows that a significant group of RNAs, mostly encoding proteins involved in synapse or cell junction formation, was shared between hippocampus and cortex (Fig 1C). In the cortex, many up-regulated RNAs encode proteins involved in RNA processing for biogenesis, while down-regulated RNAs code for proteins mediating membrane potential and synapse

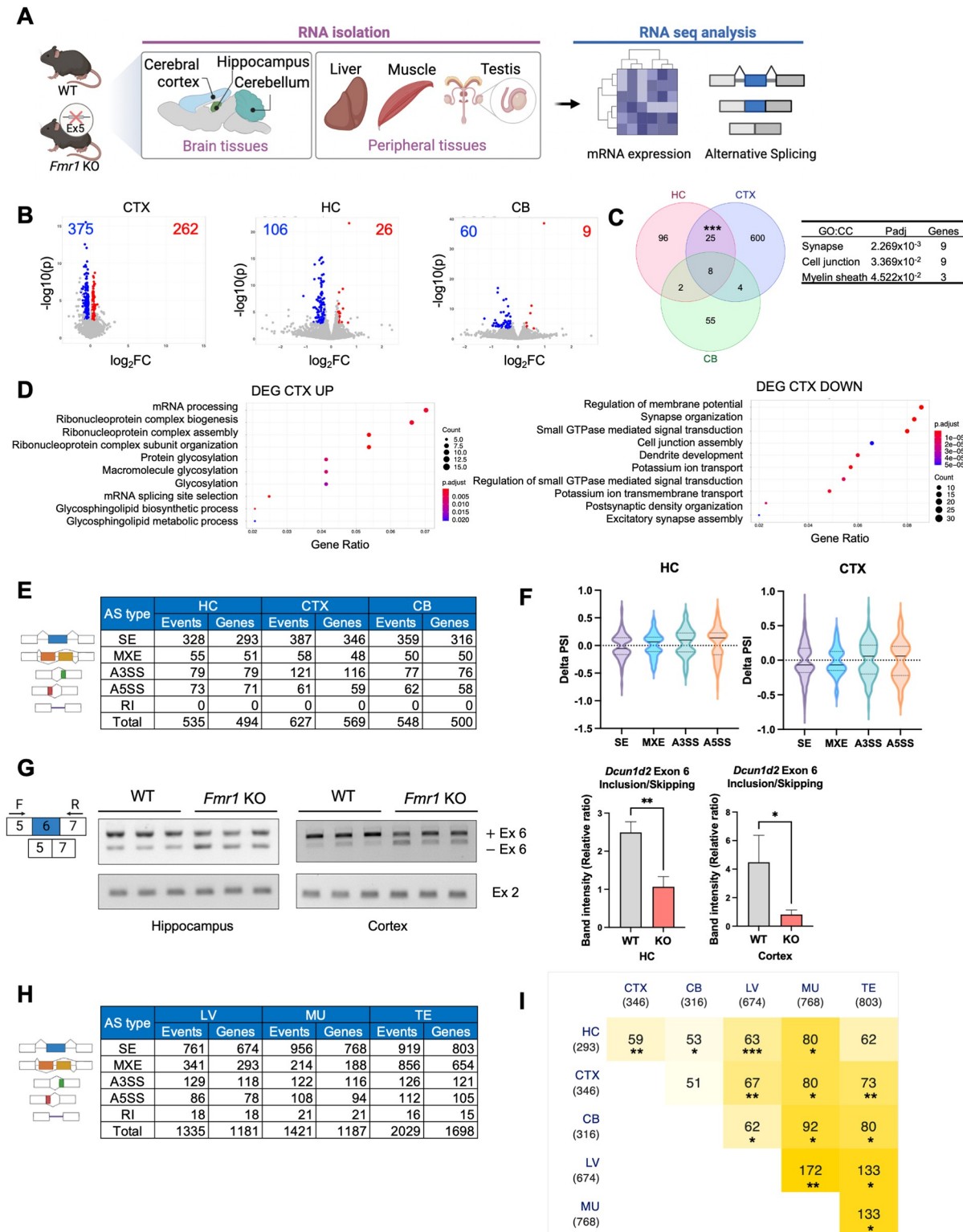

**Fig 1. Differential gene expression in *Fmr1*-deficient brain and peripheral tissues. (A)** Schematic of experiment. Created with BioRender. com. **(B)** Volcano plots of differential gene expression comparing WT and *Fmr1*-deficient cortex (CTX), hippocampus (HC), and cerebellum (CB). The numbers refer to those RNAs that are up- or down-regulated between the 2 genotypes ($n = 3$, FDR < 0.05, $\log_2$FC > 0.2 or < −0.2). **(C)** Venn diagram comparing differential RNA levels from WT and *Fmr1* KO HC, CTX, and CB (hypergeometric test, ***$p < 0.001$). GO terms for cellular components and adjusted *p*-value for overlapped RNAs are indicated. **(D)** GO terms for RNAs that are significantly up- or

down-regulated in the CTX. **(E)** Changes in alternative RNA splicing (SE, skipped exons; MXE, mutually exclusive exons; A3SS, alternative 3′ splice site; A5SS, alternative 5′ splice site; RI, retained intron) in *Fmr1* KO HC, CTX, or CB relative to WT ($n = 3$, $p < 0.05$, |delta PSI| > 0.05). **(F)** Delta percent spliced in (delta PSI) distribution for HC and CTX. The solid line is the median and the dashed lines are quartiles. *P*-value < 0.05, |delta PSI| > 0.05. **(G)** RT-PCR validation of altered *Dcun1d2* exon 6 inclusion/skipping in *Fmr1* KO HC and CTX. *Dcun1d2* constitutive exon 2 was amplified to compare total mRNA levels between the genotypes and mean ± SD is shown (Student's *t* test, *$p < 0.05$, **$p < 0.01$). **(H)** Changes in alternative RNA splicing events (SE, skipped exons; MXE, mutually exclusive exons; A3SS, alternative 3′ splice site; A5SS, alternative 5′ splice site; RI, retained intron) in *Fmr1* KO liver (LV), muscle (MU), and testis (TE) relative to WT ($n = 3$, $p < 0.05$, |delta PSI| > 0.05). **(I)** Comparison of all exon skipping changes in *Fmr1* brain regions and peripheral tissues relative to WT (hypergeometric test, *$p < 0.05$; **$p < 0.01$; ***$p < 0.001$). The underlying data can be found in S3 Data. GO, Gene Ontology; KO, knockout; WT, wild type.

organization (Fig 1D). Analysis of these RNA seq datasets demonstrates that hundreds of RNAs are mis-spliced, mostly exon skipping, in the *Fmr1*-deficient hippocampus, cortex, and cerebellum (p-value < 0.05, delta percent spliced-in [|delta PSI|] > 0.05, $n = 3$) (Fig 1E). For the hippocampus and cortex, the delta PSI has a median of about −0.06 (Fig 1F). In the cortex and hippocampus, RNAs displaying differential exon skipping between the 2 genotypes encode proteins involved in synapse organization and development and JNK signaling, respectively (S1A Fig). Fig 1G shows RT-PCR confirmation of exon 6 skipping of Defective in Cullin neddylation 1d2 (*Dcun1d2*) RNA. In both brain regions, there was a >2-fold increase in exon 6 skipping upon *Fmr1* deficiency compared to wild type (WT). Similar levels of the *Dcun1d2* constitutive exon 2 served as an internal control suggesting no change in *Dcun1d2* total RNA levels in the *Fmr1* KO tissues.

## Aberrant RNA splicing in *Fmr1* KO peripheral tissues

Because FMRP is expressed in probably all tissues, we examined RNA splicing in WT and *Fmr1* deficient liver, muscle (gastrocnemius), and testis. As with the brain, hundreds of RNAs are up- or down-regulated in FMRP KO peripheral tissues relative to WT ($\log_2$FC > 0.2 or < −0.2, FDR < 0.05, $n = 3$) (S1B Fig), which may be somewhat surprising because relative *Fmr1* levels (in transcripts per million, TPM) in these tissues are about one-tenth the amount in the brain (S1C Fig). In the liver, RNAs that are up- or down-regulated in FMRP KO relative to WT encode factors involved in various metabolic processes and catabolic and phosphorylation events, respectively (S1D Fig). In muscle, up- or down-regulated RNAs encode factors involved in extra cellular matrix organization and mitochondrial function, respectively (S1E Fig). In testis, up- or down-regulated RNAs encode factors involved in cell division and reproductive system development, respectively (S1F Fig).

In FMRP KO peripheral tissues, splicing mis-regulation is widespread; in the brain there are mostly skipped exons but many mutually exclusive exons as well ($p < 0.05$, |delta PSI| > 0.05, $n = 3$) (Fig 1H). In the liver and muscle, RNAs with differential exon skipping between the 2 genotypes encode chromatin modifying enzymes and Wnt signaling components, respectively (S1G Fig). Comparison of the RNAs from all brain regions and peripheral tissues that display significantly different exon skipping between the 2 genotypes shows a remarkable degree of overlap (Fig 1I). For example, nearly 20% of RNAs with skipped exons in hippocampus are the same as in cortex, which might be expected. However, approximately 10% of RNAs with skipped exons in the liver also exhibit exon skipping in the hippocampus. In this same vein, approximately 9% of RNAs with skipped exons in the testis also show exon skipping in the cortex. These data indicate that if FMRP regulates exon skipping in one type of tissue (e.g., the brain), it is likely to do so in another tissue (e.g., liver).

## FMRP-regulated splicing in N2A cells

To investigate the mechanism of FMRP-mediated splicing, we surmised that using a single cell type approach would be more efficacious compared to a tissue containing multiple cell types. Consequently, we used mouse N2A neuroblastoma cells depleted of *Fmr1* by an siRNA complementary to this RNA's 3′ UTR, which reduced FMRP levels by >95% compared to a nontargeting (NT) control (Fig 2A). We next performed RNA-seq from cells transfected with either the nontargeting or *Fmr1* targeting siRNAs. Figs 2B and S2A show that there were approximately 2,000 RNAs that were mis-spliced ($p < 0.05$, |delta PSI| > 0.05). Several of these mis-splicing events were validated by RT-PCR: *Mapt* (microtubule associated protein tau) exon 4, *Tnik* (TRAF2 and NCK interacting kinase) exon 21, and *Wnk1* (WNK lysine deficient protein kinase 1) exon 11 were all included more in *Fmr1*-depleted cells compared to nondepleted cells while *App* (amyloid precursor protein) exon 8, *Ski* (SKI protooncogene) exon 2, and *Os9* (osteosarcoma amplified 9, endoplasmic reticulum lectin) exon 13 were all skipped in *Fmr1*-depleted cells relative to nondepleted cells (Fig 2C).

## Rescue of mis-regulated splicing by FMRP replacement

To confirm FMRP control of splicing by an entirely different method, we used CRISPR/Cas9 gene editing to delete 7 nucleotides from exon 3 of *Fmr1*, which causes a reading frame shift to a stop codon resulting in nonsense mediated mRNA decay (Figs 2D and S2B) and a complete loss of FMRP (Fig 2E). In these KO cells, loss of *Mapt* exon 4 inclusion was nearly identical as observed with siFmr1 knockdown of *Fmr1* (Fig 2F). We next generated a reporter construct where *Mapt* exon 4 and its flanking intron sequences were inserted into the pFlareA plasmid, which contains GFP and RFP sequences. Here, if *Mapt* exon 4 is skipped, an A nucleotide will generate a start codon when juxtaposed to a TG dinucleotide following splicing to the GFP reading frame and will express GFP. If Mapt exon 4 is included, RFP will be expressed. This plasmid, together with an FMRP-expressing plasmid or an empty control plasmid, were transfected into normal or *Fmr1* KO N2A cells and green/red fluorescence intensity was analyzed by flow cytometry (Fig 2G). The quantification of mean fluorescence intensity (MFI) for both mCherry and GFP was conducted through flow cytometry analysis in both the control and *Fmr1* KO cells (Fig 2H). As depicted in Fig 2H, the inclusion of *Mapt* exon 4 was elevated in the KO in comparison to the control. Interestingly, this inclusion was reversed upon the introduction of an FMRP overexpression plasmid. Moreover, the targeted deletion of a specific MBNL1-binding site (see Fig 3 and following figures), situated near exon 4, exhibited a reduction in *Mapt* exon 4 inclusion within the *Fmr1* KO cells. Intriguingly, this binding site deletion showed no discernible effect on the control cells (S2C Fig). The western blot shows the expression level of FMRP relative to GAPDH. The "rescuing" ectopic FMRP was expressed at approximately 10% of endogenous FMRP levels. In the FMRP KO cells, *Mapt* exon 4 in the reporter was more included relative to that observed in control cells, which replicates the data with endogenous *Mapt* exon 4 with both siFmr1 depletion (Fig 2C) and CRISPR/Cas9-edited *Fmr1* KO cells (Fig 2F), albeit not to the same extent. Importantly, ectopic expression of FMRP in the KO cells restored *Mapt* exon 4 inclusion to control cells levels, demonstrating the reversibility of the exon skipping that is FMRP-dependent.

## FMRP regulation of splicing factor activity

To identify splicing factors that might be regulated by FMRP, we focused on exons in 3 RNAs that are skipped or included in *Fmr1*-deficient cells and used the SFMap database [27] to identify potential splicing factor binding sites. *Mapt* exon 4, which is more included in *Fmr1*-deficient cells relative to control cells, is flanked by binding sites for splicing factors MBNL1,

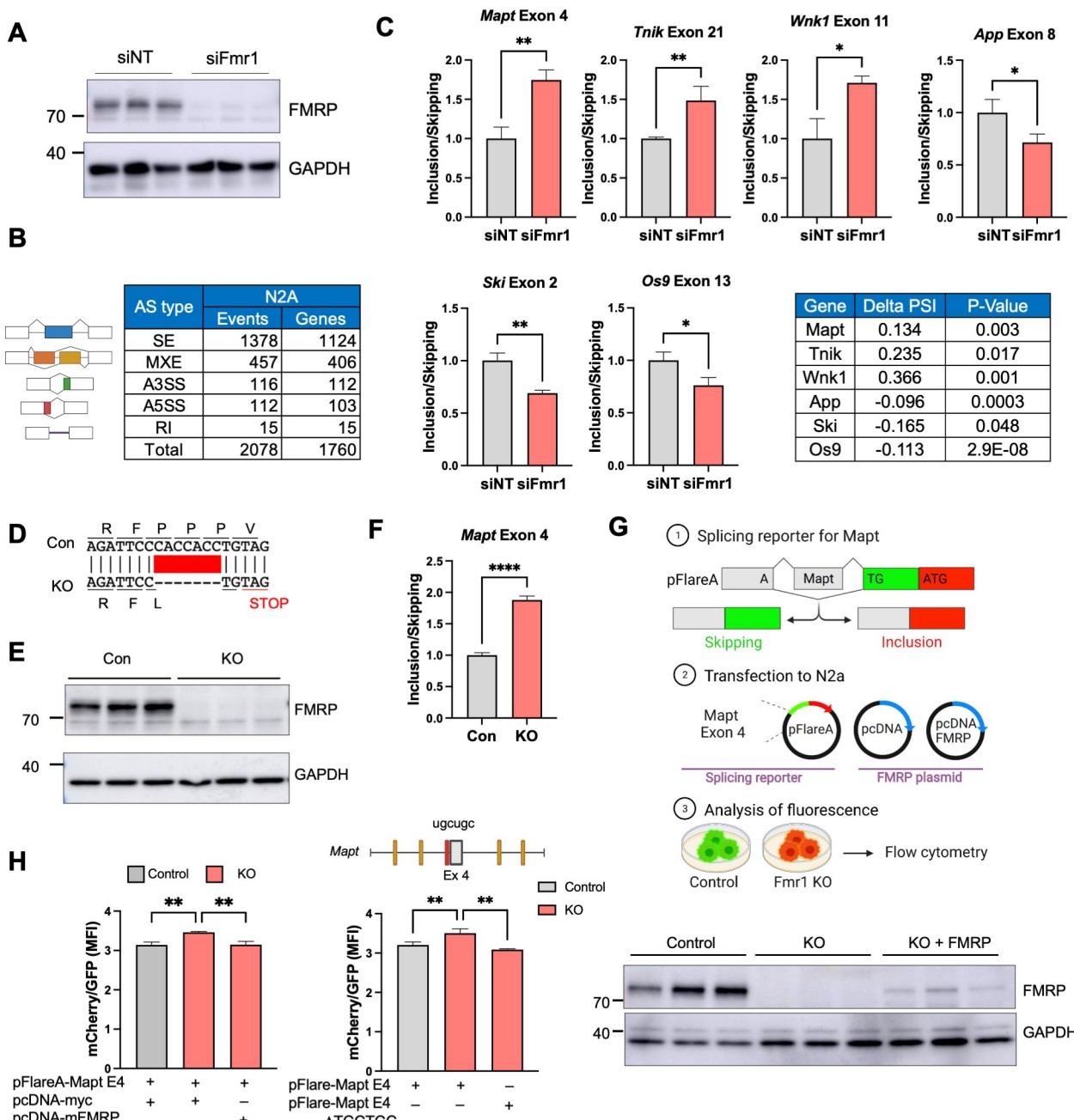

**Fig 2. Mis-regulated splicing in *Fmr1*-deficient mouse N2A neuroblastoma cells.** (A) Western blots showing depletion of FMRP following siRNA knockdown of *Fmr1*. GAPDH serves as a loading control. siNT refers to a nontargeting siRNA control. (B) Types of mis-splicing events and the number of genes affected in *Fmr1*-depleted N2A cells ($p < 0.05$, |delta PSI| > 0.05; $n = 2$ for siNT, $n = 3$ for siFmr1). (C) qPCR validation of mis-splicing events (exons skipped or included) in 6 RNAs in *Fmr1*-depleted cells compared to siNT control (Student's $t$ test, $n = 3$: *$p < 0.05$; **$p < 0.01$). Mean ± SD is shown. Delta PSI and *p*-value of selected RNAs are shown in the table. (D) CRISPR/Cas9-edited portion of *Fmr1*, which deletes 7 nucleotides leading to a frame-shift and nonsense-mediated RNA decay. (E) Western blot of FMRP in control and *Fmr1*-edited cells. (F) qPCR of *Mapt* exon 4 skipping/inclusion in *Fmr1*-edited cells compared to control and mean ± SD is shown (Student's $t$ test, ****$p < 0.0001$). (G) pFlare system for assessing exon skipping and inclusion. *Mapt* exon 4 was inserted into pFlareA. When the exon is skipped, GFP is expressed; when the exon is included, RFP is expressed. This plasmid, as well as an empty pcDNA plasmid or one that expresses mouse FMRP, was transfected into control or *Fmr1* KO N2A cells. The cells were then analyzed by flow cytometry. Created with BioRender.com. (H) pFlare splicing reporter assay. MFI of mCherry/GFP was evaluated by flow cytometry in both control and *Fmr1* KO CRISPR cell lines. At right, *Fmr1* KO CRISPR cell line was transfected with a splicing reporter that has a deletion of the MBNL1-binding site (UGCUGC) nearest to *Mapt* exon 4 (highlighted in red in the illustration). Western blot of FMRP from control cells, CRISPR/Cas9-edited cells transduced with empty pcDNA, and CRISPR/Cas9-edited cells transduced with pcDNA FMRP. The histogram quantifies the ratio of cells expressing GFP or mCherry and mean ± SD is shown (one-way ANOVA, **$p < 0.01$, $n = 3$). The underlying data can be found in S3 Data. MFI, mean fluorescence intensity.

PTBP1, hnRNPF, and hnRNPQ (Fig 3A). We depleted the RNAs encoding each of these splicing factors as well as *Fmr1* (S3A–S3H Fig). Depletion of *Mbnl1* resulted inclusion of *Mapt* exon 4 even more so compared to *Fmr1* depletion. A double depletion of both *Mbnl1* and *Fmr1* caused even greater exon inclusion than the single depletions (Fig 3B). Constitutive *Mapt* exon 15 was unaffected by these depletions (S3I Fig). Depletion of *Ptbp1* also resulted in a greater inclusion of *Mapt* exon 4 than *Fmr1* depletion. A double *Fmr1/Ptbp1* depletion was similar to *Fmr1* depletion alone (Figs 3B and S3J). Depletion of hnRNPF or hnRNPQ had no effect on *Mapt* exon 4 skipping/inclusion (Figs 3B, S3K and S3L). Because the magnitude of *Mapt* exon 4 inclusion was additive when both *Mbnl1* and *Fmr1* were depleted, we surmise that a second splicing factor under the control of FMRP is involved in this splicing event. Another possibility is that any remaining *Mbnl1* RNA, which persists after *Mbnl1* knockdown, might undergo further dysregulation following *Fmr1* knockdown.

We next examined *App* exon 8, which is also flanked by MBNL, PTBP, hnRNPF, and hnRNPQ binding sites, is skipped more frequently upon *Fmr1* depletion compared to control. *Mbnl1* depletion caused *App* exon 8 skipping at the same frequency as *Fmr1* depletion. A double depletion of *Mbnl1* and *Fmr1* was not additive for exon 8 skipping (Fig 3C). Depletion of *hnRNPF*, however, caused increased skipping of *App* exon 8 similar to that observed when *Fmr1* was depleted. A double depletion was not additive for exon skipping. *hnRNPQ* depletion did not result in any change in *App* exon 8 skipping. Depletion of these factors had little effect on skipping/inclusion of constitutive *App* exon 2 (S3M–S3P Fig).

Finally, we examined *Tnik* exon 21, which is flanked by the same splicing factor binding sites, was included more frequently when *Fmr1* is depleted (Fig 3D). While *Mbnl1* depletion had no effect on *Tnik* exon 21 skipping/inclusion, depletion of both *Ptbp1* and *hnRNPF* caused greater inclusion relative to controls (Fig 3D). Depletion of these factors had little effect on *Tnik* constitutive exon 25 (S3Q–S3T Fig). A summary of all these data demonstrates that FMRP regulation of certain splicing factors influences inclusion or skipping of specific exons (Fig 3E).

To assess whether sequences surrounding an exon regulated by FMRP are bound by these RBPs, we analyzed published CLIP-seq and RIP-seq datasets for MBNL1 and PTBP1 [28,29]. We detected MBNL1-binding sites in *Mapt* exon 4, as supported by RIP-seq data. We also observed MBNL1-binding sites downstream of *App* exon 8, as evidenced by CLIP-seq data, and similarly in *Ski* exon 2 based on a combination of RIP-seq and CLIP-seq findings (S4 Fig). We found PTBP1 binding sites, characterized by the TCTCTC/CTCTCT motif, upstream of *Mapt* exon 4, both upstream and downstream of *App* exon 8, and upstream of *Ski* exon 2. Moreover, PTBP1 binding sites were situated downstream of *Tnik* exon 21 (S5A and S5B Fig).

## FMRP regulates *Mbnl1* RNA translation

To determine whether FMRP might regulate splicing factor expression directly, we first performed RNA co-immunoprecipitation experiments followed by RT-PCR for splicing factor RNAs. Fig 4A demonstrates that FMRP co-immunoprecipitated *Mbnl1* and *Ptbp1* RNAs relative to an IgG control. For comparison, Maurin and colleagues demonstrated that mouse brain FMRP UV-CLIPs to *Mbnl1* RNA [10]. We further found that around 50% of skipped or included exons in N2A cells contain binding sites for MBNL1, while non-target exons contain binding sites at a rate of 36% (S1 and S2 Data files) using RBPmap [30].

Western blotting of the splicing factors showed that MBNL1 and hnRNPQ were elevated approximately 1.5- to 2-fold upon *Fmr1*-depletion (Fig 4B). Because neither *Mbnl1* nor *hnRNPQ* RNAs are altered by *Fmr1* depletion (S3B and S3H Fig), we infer that these 2 RNAs are under negative translational control by FMRP. MBNL1 and hnRNPQ are each represented

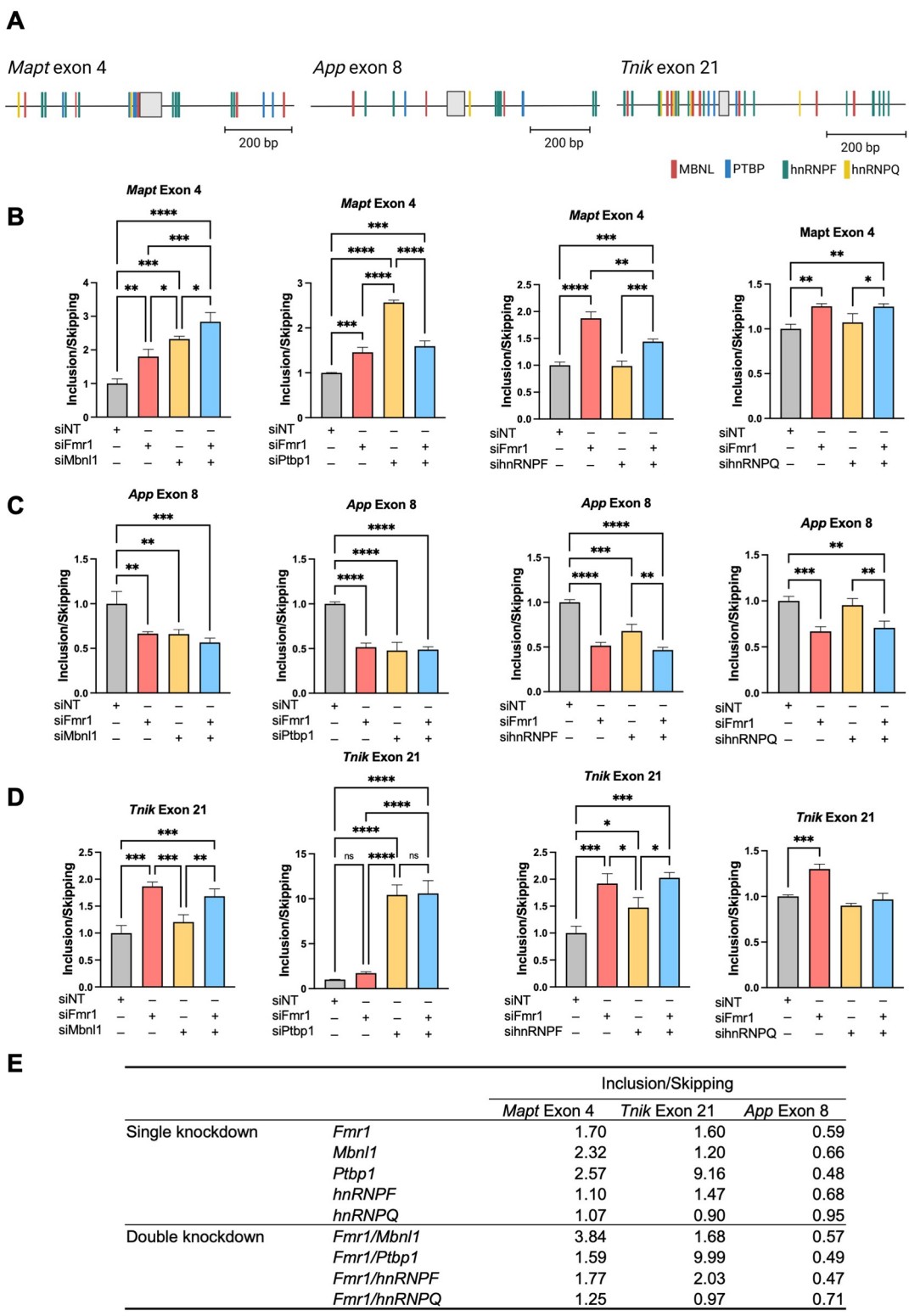

**Fig 3. RNA-binding proteins control specific splicing events in *Fmr1*-depleted N2A cells.** (A) Consensus binding motifs of MBNL, PTBP, hnRNPF, hnRNPQ, and flanking skipped or included exons of *Mapt* (exon 4), *App* (exon 8), and *Tnik* (exon 21) RNAs. Created with SFmap and Biorender.com. (B) FMRP-dependent MBNL1, PTBP1, hnRNPF, and hnRNPQ-regulated splicing of *Mapt* exon 4. All RT-qPCR determinations were made relative to GAPDH or actin (relative expression) and were performed in triplicate. *P*-values were calculated using one-way ANOVA and mean ± SD is shown. *$p < 0.05$; **$p < 0.01$;

***$p < 0.001$; ****$p < 0.0001$. (C) FMRP-dependent MBNL1, PTBP1, hnRNPF, and hnRNPQ-regulated splicing of *App* exon 8. All RT-qPCR determinations were made relative to GAPDH or actin (relative expression) and were performed in triplicate. *P*-values were calculated using one-way ANOVA and mean ± SD is shown. **$p < 0.01$; ***$p < 0.001$; ****$p < 0.0001$. (D) FMRP-dependent MBNL1, PTBP1, hnRNPF, and hnRNPQ-regulated splicing of *Tnik* exon 21. All RT-qPCR determinations were made relative to GAPDH or actin (relative expression) and were performed in triplicate. *P*-values were calculated using one-way ANOVA and mean ± SD is shown. *$p < 0.05$; **$p < 0.01$; ***$p < 0.001$; ****$p < 0.0001$. (E) Summary of exon inclusion/skipping following *Fmr1* and/or splicing factor depletion from N2A cells. The underlying data can be found in S3 Data.

by 2 isoforms; in the case of MBNL1, the slow migrating isoform is reduced when *Fmr1* is depleted while the fast migrating form is increased (Fig 4B). For hnRNPQ, the slow migrating isoform is unaffected while the fast migrating isoform is increased upon *Fmr1* depletion (Fig 4B). Neither PTBP1 nor hnRNPF undergo abundance changes in *Fmr1*-depleted cells (Fig 4B). Additional data showing that MBNL1 displays no differential stability in control versus *Fmr1*-depleted cells incubated with the proteasome inhibitors MG132 or lactacystin further indicate FMRP control of *Mbnl1* RNA translation (Fig 4C).

## FMRP regulates *Mbnl1* RNA auto-splicing and MBNL1 localization

Two of the most frequently alternatively spliced exons of *Mbnl1* mRNA are exon 5 and exon 7 (Fig 5A), of which exon 5 skipping arises by autoregulated splicing [31–33]. To determine whether alternative *Mbnl1* auto-splicing is under FMRP control and involves either of these 2 exons, we performed RT-PCR with primers that distinguish between these exons. Fig 5B shows that exon 5 is skipped more frequently upon *Fmr1* depletion while exon 7 and exon 10 (constitutive exon) skipping is unaffected.

Exon 5, which contains an NLS, determines whether MBNL1 is predominantly nuclear or is distributed to both nucleus and cytoplasm [33,34]. To assess whether exon 5 skipping upon *Fmr1* depletion alters the nucleus/cytoplasmic ratio of MBNL1, we first performed western blots of protein from cells fractionated into these 2 compartments. Fig 5C shows that the NLS-lacking MBNL1 (lower band) increased in the cytoplasm when *Fmr1* was depleted. MBNL1 containing the NLS encoded by exon 5 (i.e., the upper band) decreased in the nucleus after *Fmr1* depletion. Immunocytochemical analysis of intact cells also shows that the MBNL1 nucleus/cytoplasmic ratio decreased upon *Fmr1* depletion (Fig 5D), which is in concordance with the cell fractionation results.

FMRP shuttles to the nucleus [36] where it has been reported to co-localize with Cajal bodies [37], membrane-less structures that frequently coincide with the nucleolus. We detected a low amount of FMRP in the nucleus of N2A cells and considered that it may also associate with splicing factor-rich nuclear speckles [38]. Immunostaining for splicing factor SC-35, which detects a few splicing proteins [39], showed abundant nuclear speckles but were not co-localized with FMRP, suggesting that FMRP is unlikely to regulate splicing directly (Fig 5E).

Because we had identified a correlation between elevated SETD2, dys-regulated H3K36me3 chromatin marks, and altered splicing in *Fmr1* KO mouse hippocampus [15], we considered this might also occur in FMRP-deficient N2A cells. However, we observed no change in SETD2 levels in these cells, indicating that a changed chromatin landscape and altered splicing in FMRP-deficient cells may not be linked (S6A Fig).

## Alternative splicing of *Mbnl1* RNA in FMRP-deficient cells and tissues

We analyzed published datasets to determine whether *Mbnl1* exon skipping occurs in the FMRP-deficient tissues. Fig 6 shows that *Mbnl1* exon 5 skipping is detected not only in *Fmr1*-

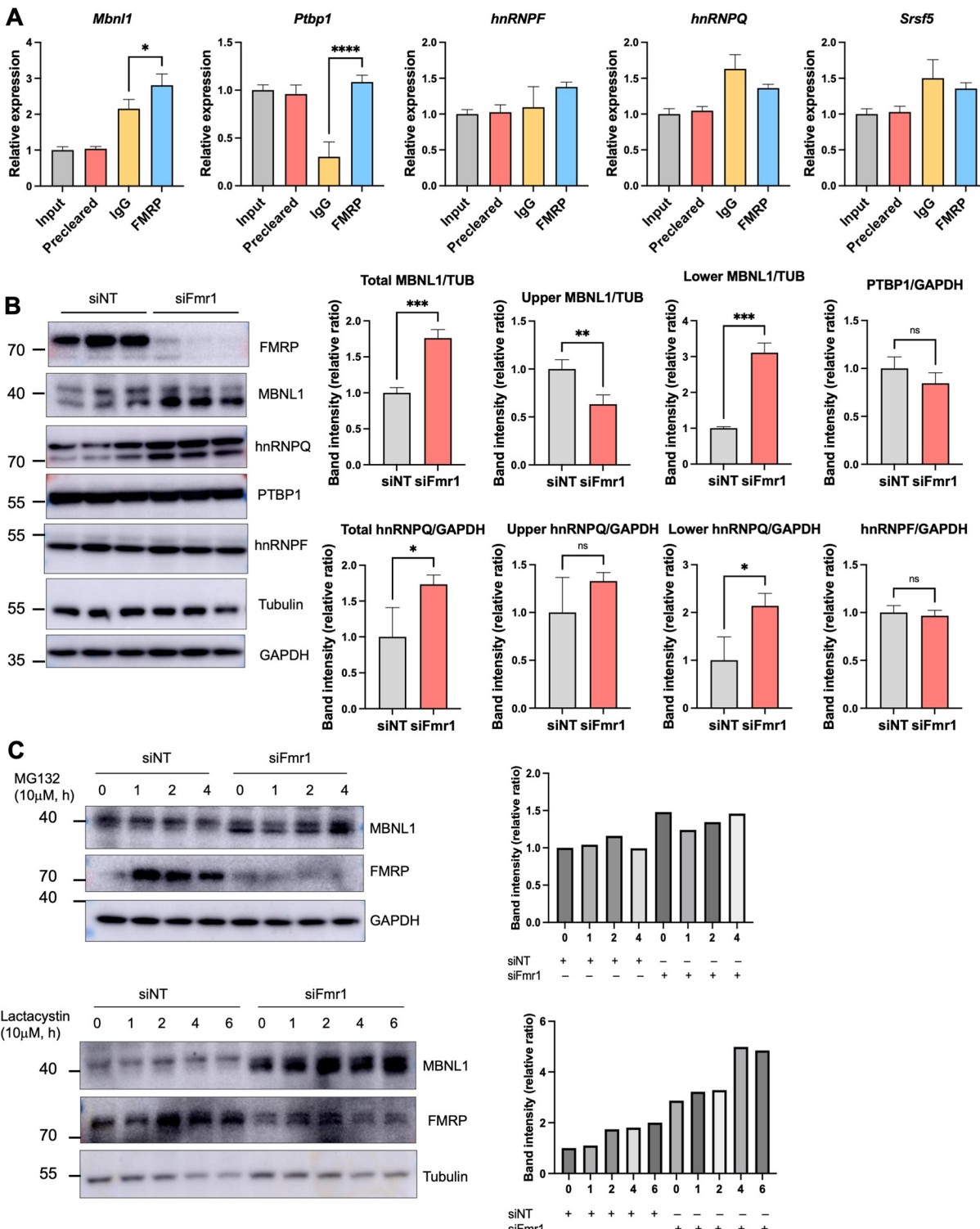

**Fig 4. FMRP regulation of *Mbnl1* RNA translation and isoform switching.** (A) Co-immunoprecipitation of *Mbnl1*, *Ptbp1*, *hnRNPF*, *hnRNPQ*, and *Srsf5* RNAs with FMRP. IgG and *Srsf5* RNA served as an immunoprecipitation controls. All experiments were performed in triplicate. *P*-values were calculated using one-way ANOVA and mean ± SD is shown. *$p < 0.05$; ****$p < 0.0001$. (B) Western blotting and quantification of splicing factors from control and *Fmr1*-depleted cells. Histogram represents band intensity quantification and mean ± SD is shown (Student's *t* test, *$p < 0.05$, **$p < 0.01$, ***$p < 0.001$). (C) Western blotting and quantification of MBNL1 in control and *Fmr1*-depleted cells following addition of the proteasome inhibitors MG132 or lactacystin for 0–4 or 6 h. The histograms represent MBNL1 band intensities relative to GAPDH or tubulin. The underlying data can be found in S3 Data.

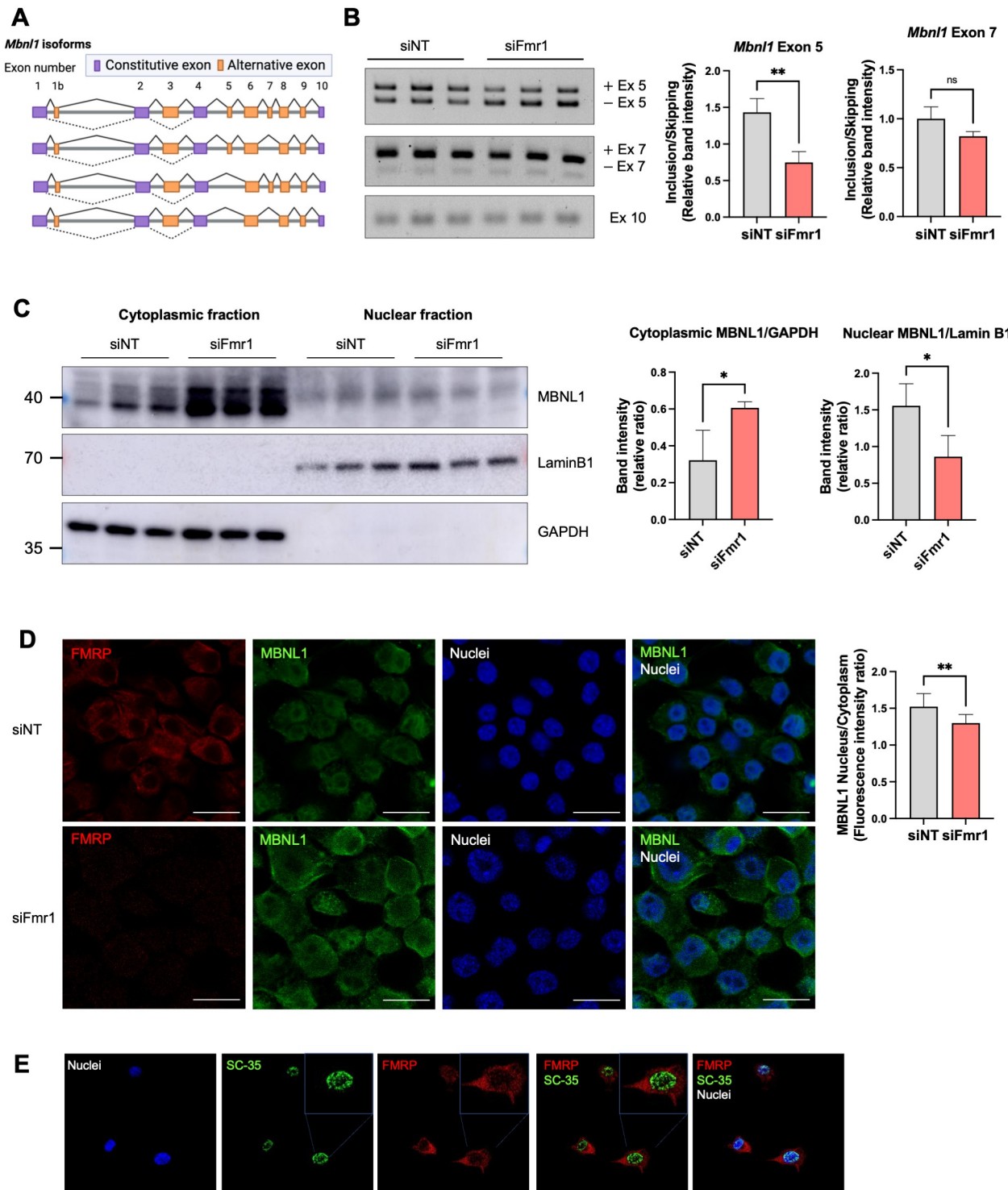

**Fig 5. FMRP control of isoform switching regulates nucleus/cytoplasmic distribution of MBNL1.** (A) Schematic illustration of *Mbnl1* isoforms (modified from Tabaglio and colleagues [35]). Exons 5 and 7 are the most frequently alternatively spliced exons. Created with Biorender.com. (B) RT-PCR of *Mbnl1* isoforms from control and *Fmr1*-depleted cell. At right is quantification of band intensities of exons 5 and 7 and mean ± SD is shown (Student's *t* test, **$p < 0.01$). The constitutive exon 10 was amplified to compare total *Mbnl1* RNA expression between the genotypes. (C) MBNL1 isoforms in the cytoplasm and nucleus in control and *Fmr1* knockdown cells. Lamin B1 and GAPDH served as makers for the nuclear and cytoplasmic fractions, respectively. Quantification of the upper and lower MBNL1 bands relative to Lamin B1 or GAPDH is indicated. Mean ± SD is shown (Student's *t* test, *$p < 0.05$). (D) Immunocytochemical localization of FMRP and MBNL1 in N2A cells following *Fmr1* depletion.

Quantification of the nucleus/cytoplasmic ratio of MBNL1 fluorescence intensity is at right. Mean ± SD is shown (Student's *t* test, **$p < 0.01$). Magnification 63×. Scale bar, 20 microns. (E) Immunocytochemistry of FMRP and SC35 in N2A cells. Magnification 63×. The underlying data can be found in S3 Data.

depleted N2A cells, but also in mouse *Fmr1* KO peripheral tissues (liver, muscle, and testis). Moreover, exon 7, which is important for MBNL1 self-dimerization, is skipped in several peripheral tissues as well as cerebellum. Although the precise function of the dimerization is unclear, exon 7 residues are thought to increase MBNL1 affinity for RNA [40]. Somewhat surprisingly, we did not detect exon 5 skipping in mouse brain, although it and exon 4 were mutually exclusive exons in human Fragile X postmortem brain. These data show that FMRP-regulated alternative splicing of *Mbnl1* is widespread, but that the exons involved in the splicing events vary according to tissue. It has been reported that specific exons are differentially alternatively spliced in various tissues due to different amounts/activities of splicing factors [41–43]. Our investigation confirms that FMRP not only influences the splicing of *Mbnl1*, but also impacts the splicing of several other RNA-binding protein mRNAs. Moreover, this effect of FMRP on splicing patterns is different across different tissue types.

## Effect of FMRP on splicing decisions by MBNL2 and PTBP2

MBNL1 and its paralog MBNL2 have the same binding sites on mRNA [28] as do the PTBP1 and its paralog PTBP2 [29]. *Mbnl2* and *Ptbp2* RNAs are present in N2A cells, but at lower levels that *Mbnl1* (by approximately 2-fold) and P*tbp2* (by approximately 8-fold) RNAs, respectively (Fig 7A). Depletion of *Fmr1* had no significant effect on *Mbnl2* RNA, but interestingly, it increased MBNL2 protein levels and induced MBNL2 auto-splicing of exon 5 (Fig 7B and S6B Fig). We found that exon 5 skipping is increased in *Fmr1*-depleted N2A cells, but that exon 5 is more included in mouse *Fmr1* KO HC (S6B Fig). Similar to *Mbnl1*, *Mbnl2* exon 5 bears an NLS, and thus determines the nuclear/cytoplasmic localization of this protein [40]. Moreover, *Mbnl2* RNA has been reported as a binding target of FMRP [10]. Depletion of *Fmr1* caused a reduction of *Ptbp2* RNA levels, but had no effect on PTBP2 protein levels (Fig 7C). Depletion of *Mbnl2* elicited an increase in *Mapt* exon 4 inclusion, significantly greater than *Mbnl1* depletion (Fig 7D). These data suggest that MBNL1 and MBNL2 may both contribute to FMRP-controlled alternative splicing, but that PTBP2 is likely to have mild effect (Fig 7E).

| Exon number | Domain, function | Species | Sample | Category | Delta PSI | P-value | Reference |
|---|---|---|---|---|---|---|---|
| Exon 1 | ZnF domain that important for RNA binding and splicing activity [33] | Mouse | LV | SE | -0.146 | 0.01 | This study |
| Exon 4/Exon 5 | ZnF domain that important for RNA binding and splicing activity/NLS [33, 44] | Human | Post-mortem cortex | MXE | 0.21 | 0.01 | [45] |
| Exon 5 | NLS [33, 44] | Mouse | LV | SE | -0.3 | 0.04 | This study |
| Exon 5 | NLS [33, 44] | Mouse | MU | SE | 0.174 | 0.0005 | This study |
| Exon 5 | NLS [33, 44] | Mouse | TE | SE | 0.191 | 0.04 | This study |
| Exon 5 | NLS [33, 44] | Mouse | N2A | SE | -0.228 | 4.54-E06 | This study |
| Exon 6 | Splicing regulatory domain, encode for bipartite NLS [33] | Mouse | Adult NSC | SE | 0.33 | 0.001 | [46] |
| Exon 6 | Splicing regulatory domain, encode for bipartite NLS [33] | Mouse | LV | A3SS | -0.082 | 0.005 | This study |
| Exon 7 | Enhances MBNL1 self-dimerization [44, 47] | Mouse | CB | SE | 0.093 | 0.03 | This study |
| Exon 7 | Enhances MBNL1 self-dimerization [44, 47] | Mouse | MU | SE | 0.044 | 0.03 | This study |
| Exon 7 | Enhances MBNL1 self-dimerization [44, 47] | Mouse | TE | SE | -0.332 | 0.0002 | This study |
| Exon 7/Exon 8 | Enhances MBNL1 self-dimerization [44, 47] | Mouse | MU | MXE | 0.051 | 0.0008 | This study |
| Exon 8 | Unknown | Mouse | MU | SE | -0.101 | 0.0001 | This study |

**Fig 6. Alternative splicing of *Mbnl1* RNA in FMRP-deficient cells and tissues.** The figure shows the tissue type and frequency where specific exons in *Mbnl1* RNA mis-spliced upon FMRP depletion [33,44–47].

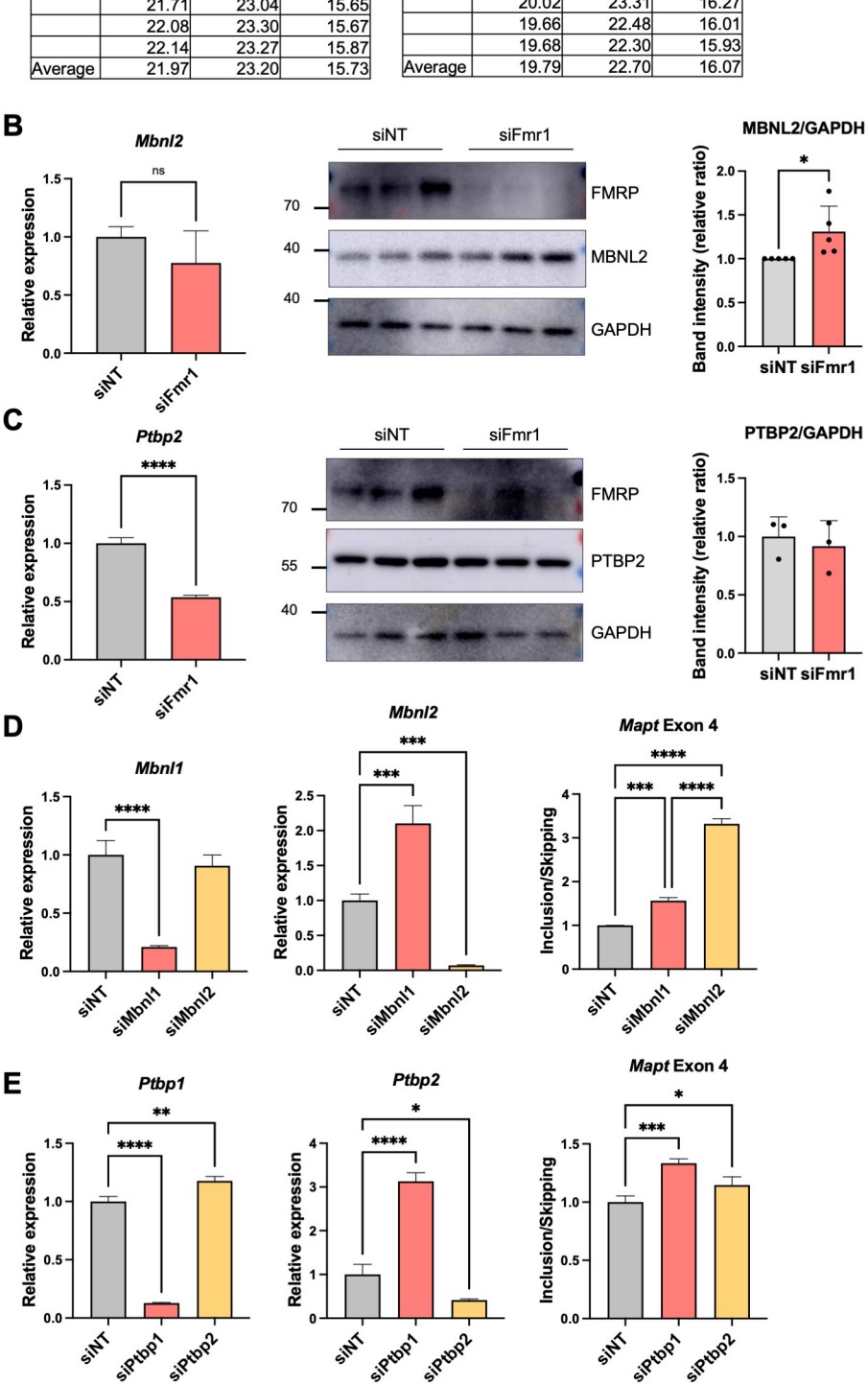

**Fig 7. MBNL2 and PTPB2 are modestly down-regulated in *Fmr1*-deficient cells.** (A) *Mbnl1*, *Mbnl2*, *Ptbp1*, and *Ptbp2* RNA levels (in Ct value) in N2A cells. (B) RT-qPCR analyses and representative western blots showing *Mbnl2* and protein levels following *Fmr1* depletion. Quantification of the MBNL2/GAPDH band intensity from 5 independent experiments is at the right. Mean ± SD is shown (Student's *t* test, *$p < 0.05$). (C) RT-qPCR analyses and western blots showing *Ptbp2* RNA and protein levels following *Fmr1* depletion. Quantification of the PTBP2/GAPDH band intensity is at the right. Mean ± SD is shown (Student's *t* test, ****$p < 0.0001$). (D) *Mbnl1* and *Mbnl2* expression

as well as *Mapt* exon 4 inclusion/skipping following *Mbnl1* or *Mbnl2* depletion. *P*-values were calculated using one-way ANOVA and mean ± SD is shown (***$p < 0.001$; ****$p < 0.0001$). (E) *Ptbp1* and *Ptbp2* expression as well as *Mapt* exon 4 inclusion/skipping following *Ptbp1* or *Ptbp2* depletion. *P*-values were calculated using one-way ANOVA and mean ± SD is shown (*$p < 0.05$, **$p < 0.01$, ***$p < 0.001$; ****$p < 0.0001$). The underlying data can be found in S3 Data.

We propose that in N2A cells, *Mbnl1* pre-mRNA undergoes alternative splicing such that exon 5-containing and exon 5-lacking mRNAs are exported to the cytoplasm where they are bound by FMRP, which limits their translation. The MBNL1 protein that retains the NLS-encoding exon 5 is transported to the nucleus where it could influence alternative splicing of other pre-mRNAs. In *Fmr1*-deficient cells, exon 5-lacking *Mbnl1* RNA is elevated in the cytoplasm relative to exon 5-containing RNA, but because there is no FMRP to limit translation in these cells, MBNL1 synthesis is robust, which is particularly the case for those *Mbnl1* mRNAs that lack exon 5 NLS. As a consequence, there is reduced MBNL1 transported to the nucleus, which may in turn have adverse effects on RNA splicing relative to normal cells. As presented in this study, MBNL1 is only one of several proteins through which FMRP regulates splicing.

## Global transcriptomic changes induced by FMRP and MBNL1

To investigate whether FMRP-mediated splicing is influenced by the absence of nuclear MBNL1, we devised an experiment involving expression of either nuclear or cytoplasmic MBNL1 isoforms. As a rescue strategy, the depletion of *Fmr1* combined with the overexpression of the nuclear MBNL1 isoform (siFmr1+nMBNL1). To identify the distinct changes that could occur from the loss of the nMBNL1, the depletion of *Mbnl1* coupled with the overexpression of the cytoplasmic MBNL1 isoform lacking exon 5 (siMbnl1+cMBNL1) (Fig 8A). Using the nMBNL1 and cMBNL1 plasmids, the localization of the expressed RNA in the nucleus and cytoplasm, respectively has been previously elucidated [48]. We validated the knockdown and overexpression of *Fmr1* and *Mbnl1* through qPCR analysis (Figs 8B and S7). It is important to note that the depletion of *Fmr1* or *Mbnl1* did not exert a significant impact on each other. As shown in Fig 8C, the overexpression of nuclear MBNL1 led to an increase in *Mbnl1* exon 5 inclusion, whereas overexpression of cytoplasmic MBNL1 resulted in an increase in exon 5 skipping.

To gain deeper insights into the global transcriptomic changes mediated by *Fmr1* and *Mbnl1*, we performed RNA-seq experiments. Consistent with the qPCR data in Fig 8B, we observed a substantial reduction in the normalized counts of *Fmr1* or *Mbnl1* in the knockdown groups and an elevation in the levels of MBNL1 isoforms in the overexpressed groups, as depicted in Fig 8D. The volcano plots provide evidence of numerous significant alterations in mRNA expression under *Fmr1* or *Mbnl1* single knockdown conditions (Fig 8E and 8F, with $\log_2 FC > 0.2$ or $< -0.2$, and FDR $< 0.05$, $n = 3$).

Strikingly, there were 660 common differentially expressed genes observed between *Fmr1* knockdown and *Mbnl1* knockdown conditions as indicated in the Venn diagram presented in Fig 8G. Of these, 377 genes exhibited up-regulation, while 236 genes were down-regulated in both experimental groups. Fig 8H highlights specific examples of these shared differentially expressed genes. For example, *Kcnab1*, encoding voltage-gated potassium channels that regulate neurotransmitter release, showed an increase in expression upon depletion of either *Fmr1* or *Mbnl1*. On the other hand, *Dr1*, encoding TBP-associated phosphoprotein that represses both basal and activated transcription levels, displayed reduced expression upon *Fmr1* depletion. Additionally, *Ccl2*, responsible for encoding monocyte chemoattractant protein-1 (MCP-1), exhibited increased expression upon *Mbnl1* depletion.

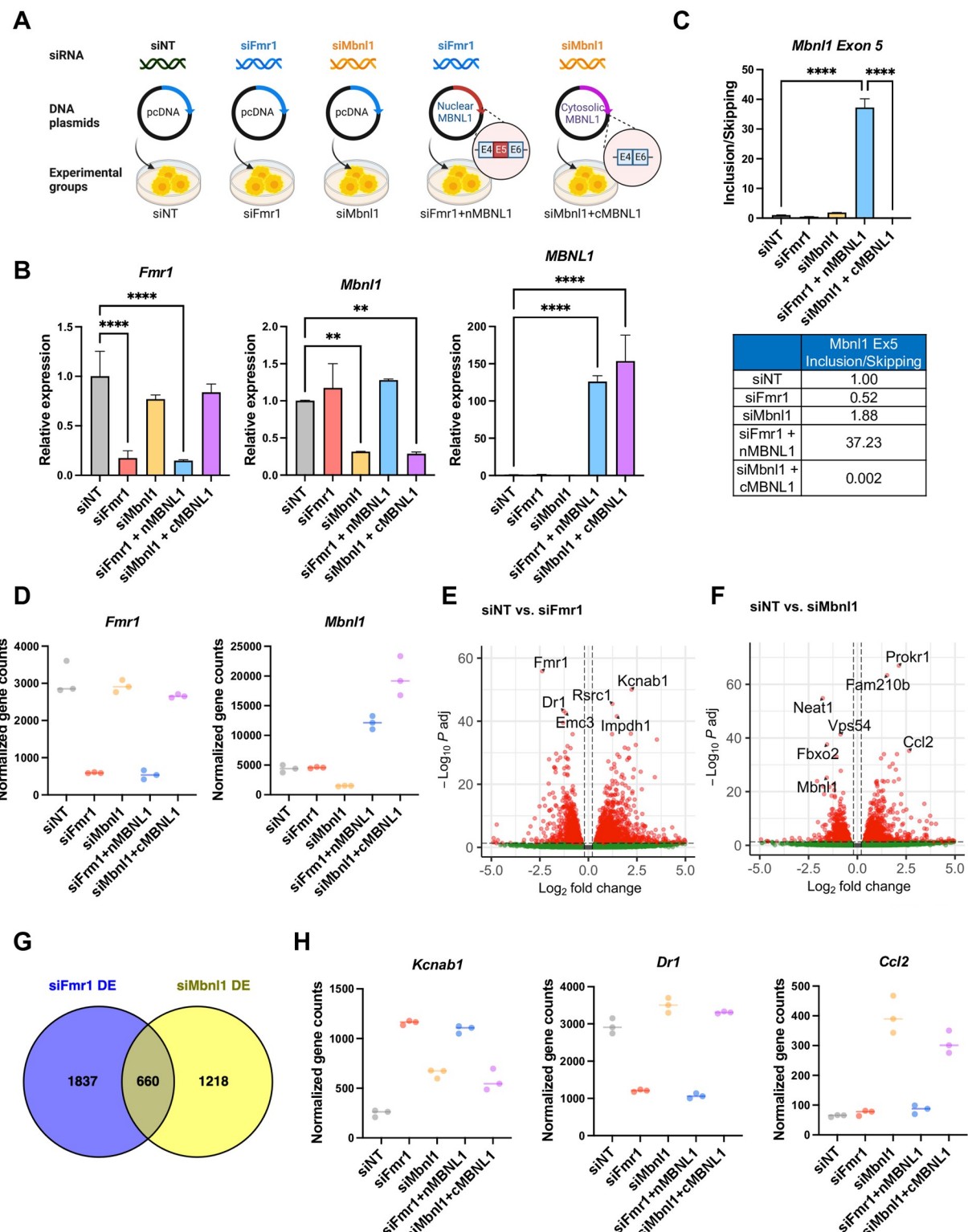

| | Mbnl1 Ex5 Inclusion/Skipping |
|---|---|
| siNT | 1.00 |
| siFmr1 | 0.52 |
| siMbnl1 | 1.88 |
| siFmr1 + nMBNL1 | 37.23 |
| siMbnl1 + cMBNL1 | 0.002 |

**Fig 8. Differential gene expression in *Fmr1* or *Mbnl1*-depleted cells.** (A) Schematic of experiment. Created with BioRender.com. (B) qPCR showing depletion of *Fmr1* following *Fmr1* or *Mbnl1* knockdown using siRNA. All RT-qPCR determinations were made relative to hprt (relative expression) and were performed in triplicate. At right, qPCR results demonstrate the overexpression of *Mbnl1*. *P*-values were calculated using one-way ANOVA and mean ± SD is shown (**$p < 0.01$; ****$p < 0.0001$). (C) qPCR of *Mbnl1* exon 5 skipping/inclusion. *P*-values were calculated using one-way ANOVA and mean ± SD is shown (****$p < 0.0001$). (D) Normalized gene counts of *Fmr1* and *Mbnl1* in the

experimental conditions. (E) Volcano plots of differential gene expression comparing siNT and siFmr1. The top 6 significantly altered genes are labeled. (F) Volcano plots of differential gene expression comparing siNT and siMbnl1. The top 6 significantly altered genes, along with *Mbnl1*, are labeled. (G) Venn diagram presenting the comparison of RNAs regulated by *Fmr1* and *Mbnl1*. (H) Normalized gene counts of representative RNAs that are regulated by *Fmr1* and/or *Mbnl1*. The underlying data can be found in S3 Data.

### FMRP-mediated SE events are regulated by nuclear MBNL1

To elucidate the role of MBNL1 in splicing events associated with FMRP, we conducted a comprehensive investigation of alternative splicing events across various groups. Our analysis revealed a substantial number of splicing alterations in *Fmr1* and *Mbnl1*-depleted cells, with notable emphasis on changes within the SE and MXE categories (Figs 9A, 9B and S8A).

We successfully validated that the skipping of *Mbnl1* exon 5 occurs upon *Fmr1* depletion and is included upon nMBNL1 overexpression. Moreover, *Mbnl1* depletion and cotransfection with cMBNL1 results in an increase in *Mbnl1* exon 5 skipping (S8B Fig). Comparing SE under both conditions, we identified an overlap of 1,458 RNAs between *Fmr1* and *Mbnl1*-depleted cells, constituting over half of each dataset (Fig 9C). Among these, 1,393 RNAs had the same alternative exon between the 2 groups (S8C Fig). Our correlation analysis based on delta PSI values indicated a robust and statistically significant positive correlation between *Fmr1* and *Mbnl1*-depleted cells, with a correlation coefficient (r) of 0.8382, R-squared ($R^2$) value of 0.7026, and *p*-value of less than 0.0001 (Fig 9D).

We next examined whether splicing could be rescued by overexpression of nuclear MBNL1, which includes exon 5, in *Fmr1*-depleted cells. Remarkably, we found that 21% of mis-spliced exons in *Fmr1*-depleted cells were restored by nuclear MBNL1 (Figs 9E and S9A). We extended our inquiry to include a comparison of splicing events in *Fmr1*-depleted cells and cytoplasmic MBNL1 overexpressed cells, which had a reduced level of endogenous *Mbnl1*. Notably, 37% of mis-spliced exons in *Fmr1*-depleted cells exhibited congruent splicing patterns with those in the siMbnl1+cMBNL1 group (Fig 9F). Specific examples include *Slc30a4* exon 2 and *Carm1* exon 5, where *Fmr1* knockdown disrupted splicing, but nMBNL1 overexpression reversed this effect (Figs 9G, 9H and S9B).

Furthermore, we uncovered splicing factors whose mRNA expression or splicing patterns were altered exclusively in *Fmr1*-depleted cells and not in *Mbnl1*-depleted cells. This observation suggests the existence of non-MBNL1-mediated mechanisms contributing to splicing regulation under *Fmr1-depletion* (S9C and S9D Fig).

## Discussion

The proteome of the hippocampus, an exceptionally well-studied brain region of FXS model mice, is largely attributed to altered mRNA translation with perhaps a minor contribution of protein degradation [3,49–51]. This study indicates that mis-regulated alternative splicing may be a contributor to the Fragile X proteome not only in the hippocampus and other brain regions of *Fmr1*-deficient mice, but in peripheral tissues as well. Our investigation of the mechanism of FMRP-mediated splicing used *Fmr1*-deficient N2A cells, which was based on the assumption that a single cell type would more likely reveal the involvement of specific factors than a complex mixture of cells such as in the brain. By mapping splicing factor binding sites flanking certain skipped or included exons in 3 mRNAs in *Fmr1*-depleted cells, we found that 4 proteins: MBNL1/2, PTBP1, and hnRNPF contribute to alternative splicing mis-regulation, and MBNL1/2 and hnRNPQ, are translationally inhibited by FMRP. Moreover, *Mbnl1/2* auto-splicing induced skipping of the NLS-containing exon 5, which is thought to be enhanced by elevated levels of MBNL1/2 protein [40,47], was observed. This event impairs MBNL1/2 nuclear transport, which in turn likely affects downstream splicing decisions. AS events altered

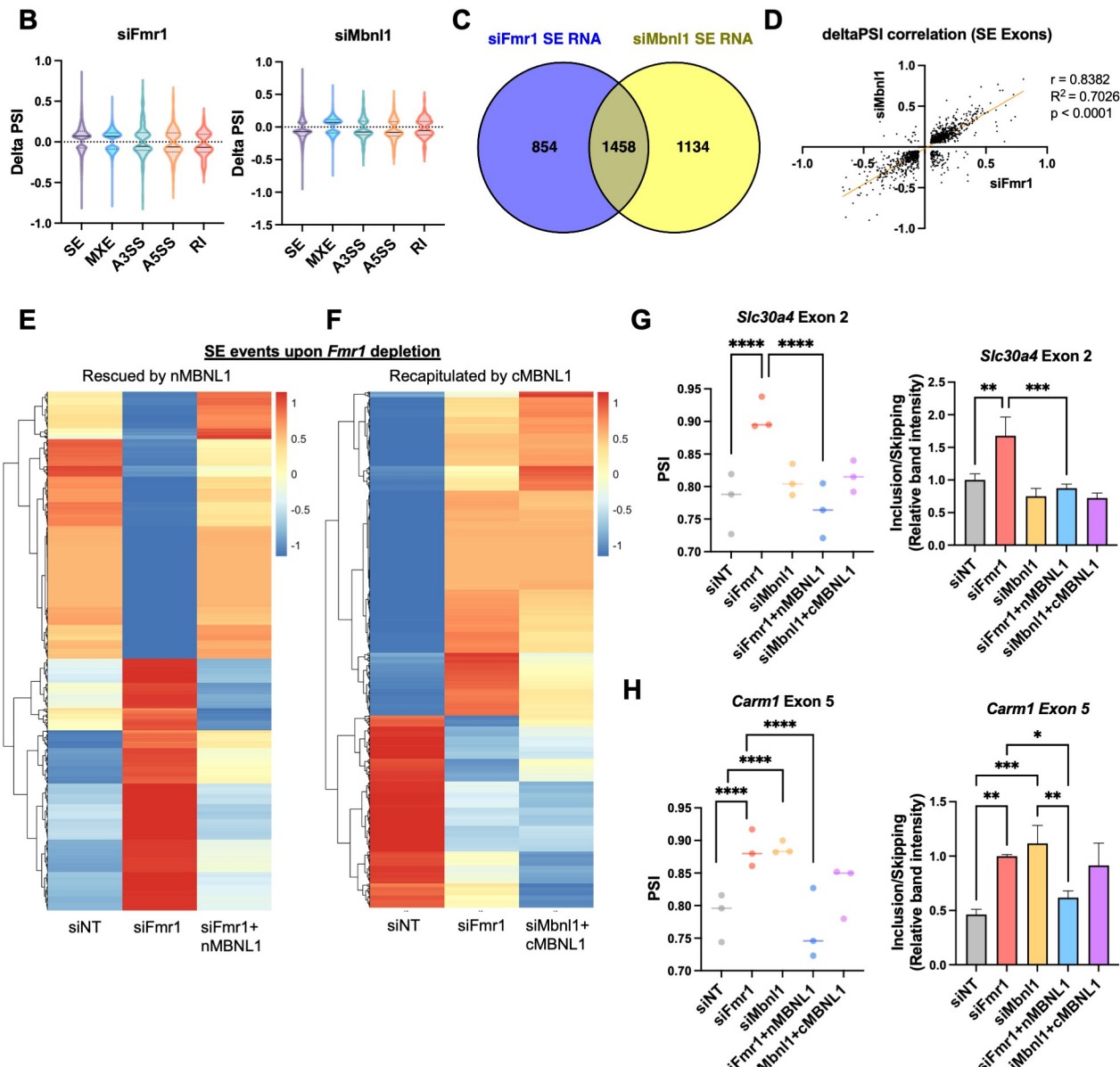

**Fig 9. FMRP regulates alternative splicing through redistribution of MBNL1 isoforms.** (A) Changes in alternative RNA splicing across categories. (B) Violin plots illustrating the distribution of delta PSI in *Fmr1* and *Mbnl1*-depleted cells. The solid line is the median and the dashed lines are quartiles. *P*-value < 0.05, |delta PSI| > 0.05. (C) Venn diagram comparing RNAs exhibiting skipped exon events in *Fmr1* and *Mbnl1*-depleted cells. (D) Correlation plot showing the relationship of delta PSI values between *Fmr1* and *Mbnl1*-depleted cells. (E) Heatmap visualizing the PSI values of mis-spliced RNAs in *Fmr1*-depleted cells, rescued by the overexpression of nMBNL1. PSI values were adjusted by z-score. (F) Heatmap displaying the PSI values of mis-spliced RNAs in *Fmr1*-depleted cells and cytoplasmic MBNL1 expressing cells (siMbnl1+cMBNL1), which is similar to the splicing changes upon *Fmr1*-depletion. PSI values were adjusted by z-score. (G) PSI values of *Slc30a4* exon 2, rescued by nMBNL1. *P*-values were calculated using rMATS (****p < 0.0001). At right, RT-PCR validation of *Slc30a4* exon 2 splicing. Band intensity was quantified and mean ± SD is

shown (ANOVA, **$p < 0.01$, ***$p < 0.001$). (H) PSI values presented for *Carm1* exon 5. *P*-values were calculated using rMATS (****$p < 0.0001$). At right, RT-PCR validation of *Carm1* exon 5 splicing. Band intensities were quantified and mean ± SD is shown (ANOVA, *$p < 0.05$, **$p < 0.01$, ***$p < 0.001$). The underlying data can be found in S3 Data.

by FMRP and MBNL1 demonstrate a robust positive correlation. Moreover, the ectopic expression of MBNL1 isoform containing the NLS within exon 5 reversed approximately one-fifth of the disrupted splicing pattern in *Fmr1*-depleted cells. Conversely, expression of mainly the cytoplasmic MBNL1 recapitulated a proportion of the splicing changes observed upon *Fmr1* depletion. In summary, our collective findings underscore the existence of discrete subsets of nuclear MBNL1-mediated splicing events within the context of *Fmr1*-regulated splicing.

*Mbnl1* exon 5 is also skipped in *Fmr1*-deficient mouse peripheral tissues as well as in human postmortem Fragile X brain. Exons 6, 7, and 8 are skipped in neural stem cells, and/or liver, muscle, testis, and cerebellum from *Fmr1*-deficient mice. Thus, FMRP regulation of *Mbnl1* splicing is complex and is strongly influenced by cell/tissue-type, which likely contributes to downstream splicing regulation.

The regulation of splicing via MBNL1/2 is only one of several FMRP-dependent mechanisms that mediate RNA processing. PTBP1 and hnRNPF all influence splicing decisions that are downstream of FMRP. For both MBNL1 and hnRNPQ, this involves FMRP-regulated translation of their respective mRNAs. In this sense, FMRP control of splicing is similar to FMRP control of chromatin modifications and transcription; the root cause of the alteration of these molecular events is dys-regulated translation when FMRP is absent [15,19]. We also considered whether FMRP might influence splicing directly. It is a nuclear shuttling protein that at least in mouse testis, binds chromatin and is involved in the DNA damage response [52]. FMRP co-localizes with Cajal bodies in Hela cells, which implies it may modify rRNA biogenesis [37]. We inferred that if FMRP was a direct regulator of splicing, it would co-localize with SC35-containing nuclear splicing/processing bodies or speckles [38]. We did not detect any such co-localization and thus FMRP is unlikely to be a direct modulator of splicing. In addition, we previously reported a correlation between the up-regulation of SETD2, altered H3K36me3 chromatin marks, and RNA splicing mis-regulation in *Fmr1*-deficient mouse brain [15]. In *Fmr1* KO N2A cells, however, we detected no alteration in SETD2 levels, and thus a change in H3K36me3 leading to splicing dys-regulation is unlikely. The notable disparity observed between the brain and the cellular model serves to highlight the intricacies of molecular regulation and the intricate manner in which FMRP-mediated processes operate. The multifaceted interplay involving FMRP, SETD2, splicing factors, and the dysregulation of splicing emphasizes the need for a more comprehensive investigation into the mechanisms upon the specific cellular context.

In most cases, the dys-regulated inclusion/exclusion of exons in *Fmr1*-deficient tissues/cells has a mean of approximately 20%, but with a large distribution. Although the magnitude of such changes is within the range often observed for alternative splicing [53], it is unclear to what extent these splicing changes have biological consequences. However, even modest changes in exon skipping can manifest themselves with changes in biology if a skipped exon is regulatory. For example, an exon encoding a regulatory phosphorylation site in the RNA-binding protein CPEB4 is skipped <30% of the time but this skipping is correlated with if not causative for autism [54]. In the *Fmr1* KO mouse, we cannot ascribe any single mis-splicing event as contributing to a Fragile X phenotype. Instead, it is more likely that the amalgamation of hundreds of mis-splicing events results in some Fragile X pathophysiology, for example, dys-regulated synaptic transmission or learning and memory [7,55].

Finally, the dys-regulated splicing in Fragile X model mice may represent a point of convergence with other neurodevelopmental disorders [56]. For example, splicing is impaired in autism spectrum disorders [57], Rett Syndrome [58], Pten [59], and others [15]. Whether mis-splicing in these disorders are related mechanistically is unclear, but they may involve several of the same factors (e.g., MBNL1, PTBP1). More intriguing is the prospect that some mis-splicing events link similar behavioral or other physiological impairments among these disorders. This may especially be the case when very small microexons encoding regulatory domains are skipped [60]. Future studies will be necessary determine whether specific mis-splicing events promote pathophysiolgical outcomes.

## Materials and methods

### Ethics statement

Animal maintenance and experimental procedures were performed as approved by the University of Massachusetts Chan Medical School Institutional Animal Care and Use Committee (IACUC, PROTO201900332).

### Animals

Mice were housed under a 12 h light/dark cycle with free access to food and water. Wild-type and *Fmr1* KO mice were purchased from the Jackson Laboratories. Two- to 3-month-old male mice were used in this study (*n* = 3 each for WT and *Fmr1* KO).

### Cell culture and siRNA transfection

N2A and HEK293T cells were cultured in Dulbecco's Modified Eagles Medium (DMEM) supplemented with 10% fetal bovine serum (FBS) and antibiotics. The siRNAs targeting *Fmr1*, *Ptbp1*, *Ptbp2*, *hnRNPF*, *hnRNPQ*, *Mbnl1*, and *Mbnl2* were purchased from IDT. As a negative control siRNA, siNT (ON-TARGETplus) was purchased from Dharmacon. For siRNA transfection, $1 \times 10^5$ cells were seeded in 6-well plates overnight and transfected with 20 to 25 pmol of the indicated siRNAs using Lipofectamine 3000 (Thermo Fisher Scientific, 13778030) following the manufacturer's instructions. For double depletion experiments, cells were transfected in triplicate as follows: (1) siNT (80 pmol); (2) siFmr1 (40 pmol) + siNT (40 pmol); (3) siMbnl1 (40 pmol) + siNT (40 pmol); and (4) siFmr1 (40 pmol) + siMbnl1 (40 pmol). Cells were incubated with the indicated siRNAs for 48 to 72 h before being analyzed. For co-transfection of siRNA and DNA, N2A cells were transfected with 25 pmol of siRNA and 1 μg of DNA with 7.5 μl of Lipofectamine 3000. Cells were collected 72 h after transfection.

### RT-PCR and RT-qPCR

Total RNA was isolated using TRIzol-LS (Invitrogen, 10296–028) and total RNA (1 μg) was reverse transcribed using QuantiTect Reverse Transcription Kit (Qiagen, 205313) according to manufacturer's instructions. RT-PCR was performed using GoTaq Green Master Mix (Promega, M7123). Approximately 2 μl of diluted cDNA was added to 12.5 μl GoTaq Green Master Mix and 0.4 μm of forward and reverse primers and nuclease-free water in a 25 μl reaction. PCR amplification was performed as follow: initial denaturation at 95˚C for 2 min, 30 cycles of denaturation at 95˚C for 30 s, annealing at each primer's annealing temperature for 1 min, and 72˚C for 1 min/kb and final extension at 72˚C for 5 min. qPCR was performed with QuantStudio3 (Thermo Fisher Scientific) as follow: initial denaturation at 95˚C for 10 min, 39 cycles of denaturation at 95˚C for 15 s annealing, and extension at 60˚C for 1 min. For alternative splicing validation, primers were designed to specifically amplify exon–exon junctions of the

included or skipped isoform. A primer pair amplifying a constitutive exon in each mRNA was used to determine changes in total mRNA expression between genotypes. Primer sequence information is listed below.

## Western blot

Cells were washed with ice-cold PBS and collected using trypsin. After centrifugation, the cells were lysed with ice-cold RIPA buffer (150 mM NaCl, 1% Triton X-100, 0.5% sodium deoxycholate, 0.1% SDS, and 50 mM Tris-HCl (pH 8.0)) with cOmplete Mini EDTA-free protease inhibitor cocktail (MilliporeSigma, 11836170001) and PhosSTOP (MilliporeSigma, 4906837001) and rotated for 10 min at 4˚C. The lysates were collected by centrifugation at 12,000 rpm for 10 min at 4˚C. Supernatants were removed and the protein concentration was quantified using the colorimetric assay by Pierce BCA protein assay kit (Thermo Fisher Scientific, 23225). Protein lysates were resolved using 10% SDS-PAGE gels and transferred to 0.45 μm PVDF membranes (Millipore, IPVH00010). The membranes were blocked in 5% skim milk solution for 1 h at RT, then incubated with primary antibody at 4˚C overnight: anti-FMRP antibody (Abcam, ab17722, 1:1,000), anti-GAPDH antibody (Cell signaling technology, 2118, 1:1,000), anti-alpha-Tubulin (MilliporeSigma, T5168, 1:1,000), anti-Lamin B1(Abcam, ab16048, 1:2,000), anti-MBNL1 (Cell signaling technology, 94633, 1:1,000), anti-MBNL2 (Cell signaling technology, 93182, 1:1000), anti-PTBP1 (Cell signaling technology, 57246, 1:1,000), anti-PTBP2 (Cell signaling technology, 59753, 1:1,000), anti-hnRNPF (Novusbio, NBP2-57442-25 μl, 1:1,000), anti-hnRNPQ (Abclonal, A9609, 1:1,000), anti-SETD2 (Abclonal, A11271, 1:1,000), anti-histone H3K36me3 (Abcam, ab9050, 1:1,000), anti-histone H3 (Abcam, ab18521, 1:1,000). The membranes were incubated with horse radish peroxidase (HRP)-linked secondary anti-rabbit (Jackson ImmunoResearch, 211-032-171, 1:5,000) or anti-mouse (Jackson ImmunoResearch, 115-035-174, 1:5,000) antibody and developed with ECL (Pierce, NEL105001EA). Immunoreactive bands were detected using GE Amersham Imager.

## Cytosol and nuclear protein fractionation

Cells were washed with ice-cold PBS, collected by trypsinization, pellets collected by centrifugation, and then resuspended in Triton extraction buffer (TEB, PBS containing 0.5% triton X-100 (v/v), 2 mM phenylmethylsulfonyl fluoride, 0.02% NaN$_3$) and lysed on ice for 10 min. Following a centrifugation at 12,000 rpm at 4˚C, the supernatants were saved for cytoplasmic protein and the pellets were resuspended in nuclear lysis buffer (50 mM Tris-HCl (pH 7.4), 120 mM NaCl, 1 mM EDTA, 1% Triton X-100, 0.1% SDS) and lysed by sonication at high power for 8 cycles (15 s on, 60 s off) using a Bioruptor (Diagenode). The lysates were collected after centrifugation at 13,000 rpm for 10 min at 4˚C and the supernatants were prepared for nuclear protein analysis. Nuclear and cytoplasmic protein concentrations were measured using BCA assays.

## RNA-seq

Mouse tissues were powdered in liquid nitrogen with a frozen mortar and pestle. For RNA extraction, TRIzol was added to the tissue powder and homogenized with Dounce tissue homogenizer. The RNA was treated with TurboDNase (Invitrogen, AM2238) to remove genomic DNA contamination. For peripheral tissues and N2A cells, total RNA was extracted, and the integrity analyzed by a fragment analyzer. Library preparation and RNA sequencing were performed by Novogene (California, USA) with NovaSeq 6000 sequencer (150 base paired ends). For brain samples, polyadenylated mRNA was enriched using Nextflex Poly(A) Beads (NEXTflex, Bioo Scientific Corp, 512980) and cDNA libraries were prepared using a NEXTflex

Rapid Directional qRNA-Seq Kit (Bioo Scientific Corp, NOVA-5130-03D). In brief, the mRNA was randomly fragmented, reverse transcribed, and double-stranded cDNA was adenylated and ligated to index adapters. The cDNA was amplified by PCR and purified with AMPure beads (Beckman Coulter, A63881). The libraries were quantified with a KAPA Library Quantification Kit (KAPA Biosystems, KK4873) and the quality and size were analyzed by a fragment analyzer. Pooled libraries were sequenced on an NextSeq500 Sequencer using NextSeq 500/550 High Output Kit v2.5 (Illumina, 20024906, 75 base paired ends).

## Differential expression and alternative splicing analysis

RNA-seq analysis was performed using DolphinNext pipeline at UMass Chan Medical School [61] or as described. Quality trimming was conducted using Fastqc (v0.11.8) and Trimmomatic (v.0.39). Reads below a minimum quality PHRED score of 15 at the 10 nt sliding window were first clipped and the clipped reads shorter than 25 nt were trimmed. The trimmed reads were mapped to rRNA by Bowtie2 (v2.3.5) were further filtered out. The cleaned reads were aligned to the mouse reference genome (mm10) with STAR (v1.16.1), and gene expression was quantified by RSEM (v1.3.1). Differential gene expression was analyzed using DESeq2 (v1.16.1). The FDR adjusted $p$-value $< 0.05$ and $\log_2 FC > 0.2$ or $< -0.2$ was used as the cut-offs to identify the differentially expressed genes. Alternative splicing events are analyzed using rMATS (v3.0.9) [62] and $p$-value $< 0.05$ and $|\text{delta PSI}| > 0.05$ was used as the cut-offs for splicing events. To assess biological function, Gene Ontology (GO) term analysis was conducted using clusterProfiler R package [63,64]. Significant RNA Overlap from WT and *Fmr1* KO tissues was analyzed using DynaVenn [65] using $p$-value ordered RNA list.

## Generation of an *Fmr1* CRISPR/Cas9-edited cell line

To construct an *Fmr1* KO N2A mouse cell line, an *Fmr1* exon 3 DNA oligonucleotide was inserted into pLentiCRISPR v2 (Addgene, 52961) adapted from published methods [12]. Briefly, annealed and phosphorylated oligonucleotides were cloned into a FastDigest BmsBI (Fermentas)-digested vector following the manufacturer's protocol. pLentiCRISPR-*Fmr1* Exon3 was co-transfected with pMD2.G and psPAX2 into HEK293T cells. The viral particles containing supernatants were collected after 48 h of transfection by filtering through 0.45 μm filters and transduced to N2A cells. After 3 days of infection, transduced cells were selected with puromycin for 2 weeks. Puromycin-resistant cells were seeded in each well of a 96-well plate with a single cell per well. Single cell-derived colonies were obtained after several weeks of culture and verified for *Fmr1* knockout by Sanger DNA sequencing and western blotting. For the sequencing, genomic DNA was extracted using lysis buffer (10 mM Tris 8.0, 200 mM NaCl, 20 mM EDTA, 0.2% Triton X-100 and 100 μg/ml proteinase K) and the deleted exon region was PCR amplified using primers (sequences noted below). To identify deleted sequences, the PCR products were cloned with a TOPO TA Cloning Kit (Thermo Fisher Scientific, 450030) followed by sequencing using T7 primers (Genewiz).

## Alternative splicing reporter system

To generate an alternative splicing reporter, total DNA was isolated from N2A cells using the lysis buffer described above. *Mapt* exon 4 and flanking the intron regions were PCR amplified using Phusion High-Fidelity DNA polymerase and inserted into NheI/BamHI digested pFlareA Plasmid (Addgene, 90249) and sequenced. For MBNL1-binding site deletion mutant, the MBNL1-binding site (UGCUGC) was deleted using Q5 Site Directed Mutagenesis kit (NEB E0554), and 12.5 ng of pFlareA-Mapt Ex4 splicing reporter was used as template and

mixed with 12.5 µl of Q5 Hot Start High-Fidelity 2X Master Mix, 2.5 µl of 10 µM of each primer (Forward: CTCGGACCAGCCGAAGAA, Reverse: AAGGGAGAGGACAGAAGG), and 9 µl of nuclease free water. PCR amplification was performed with initial denaturation at 98˚C for 30 s, 25 cycles of denaturation at 98˚C for 10 s, annealing at 63˚C for 30 s, extension at 72˚C for 3 min 30 s, and final extension at 72˚C for 5 min. PCR product was treated with kinase, ligase, and DpnI according to manufacturer's instruction. Deletion of MBNL1-binding site was confirmed by Sanger sequencing.

Cultured N2A control and *Fmr1* CRISPR/Cas9 KO cells were seeded in 6-well plates overnight and then transfected using 7.5 µl of Lipofectamine 3000 (Invitrogen) and 5 µl of P3000 with the 1 µg of pFlareA-Mapt exon4 splicing reporter. For the rescue experiment, 1.5 µg of pcDNA-myc or pcDNA-mouse FMRP ectopic expression plasmids was added. Transfected cells were washed with PBS and collected by trypsinization 48 h after transfection. GFP and mCherry fluorescence intensities were detected using flow cytometry (LSR II A-5 Donald).

## RNA-immunoprecipitation (RNA-IP)

N2A cells were transfected with siNT and siFmr1 using Lipofectamine 3000. After 72 h of incubation, the cells were washed with fresh media containing 100 µg/ml cycloheximide (CHX, MilliporeSigma, C4859). After washing with ice-cold PBS-containing CHX, the cells were pelleted and lysed in 1× polysome buffer (20 mM Tris-HCl (pH 7.5), 5 mM MgCl₂, 100 mM KCl, 1 mM DTT, 100 µg/ml CHX, protease inhibitor cocktails, 1% Triton X-100 (v/v)) with 10 passages through a 25 G needle to triturate and incubated on ice for 10 min. The lysates were centrifuged at $14,000 \times g$ at 4˚C and RNA concentration was measured using Qubit BR RNA Assay Kits (Thermo Fisher Scientific, Q10210). For IP, 5 µg of RNA was pre-cleared with 25 µl of Protein G Dynabeads (Invitrogen, 10003D) for 30 min at 4˚C. Approximately 10% of aliquot of the precleared lysates were saved as an input, and 2.5 µg of FMRP antibody (Abcam, ab17722) or IgG (MilliporeSigma, 12–370) was added to the precleared lysates and incubated for 2 h at 4˚C. A total of 25 µl of Protein G Dynabeads was added and incubated for 30 min at 4˚C and the beads were gently washed with wash buffer (20 mM Tris-HCl, 100 mM KCl, 5 mM MgCl₂, 1% Triton X-100 (v/v)) for 3 times. RNA was extracted using TRIzol and 100 ng of RNA were reverse transcribed using Quantitect followed by qPCR using iTaq SYBRgreen (Bio-rad, 1725122).

## Protein stability assay

N2A cells were transfected with siNT and siFmr1 using Lipofectamine 3000. After 72 h of incubation, cells were treated with 10 µM MG132 (MilliporeSigma, 474790) or lactacystin (Tocris, 2267) and harvested at different time points.

## Immunocytochemistry

For immunofluorescent staining, $1 \times 10^5$ cells were seeded in a Chamber Slide (Nunc Lab-Tek II CC2, 154917) and transfected with 10 pmol siNT and siFmr1. After 48 h, cells were washed and fixed with 4% formaldehyde solution (Thermo Fisher Scientific, AAJ19943K2) for 10 min at RT. The fixed cells were washed with PBS 3 times and permeabilized using 0.1% Triton X-100 in PBS for 15 min at RT. The cells were washed with PBS 3 times and incubated with blocking buffer (1% BSA in PBS) for 1 h at RT. Cells were then incubated overnight at 4˚C with primary antibodies anti-FMRP (Abcam, Ab17722, 1:500), anti-MBNL1 (Thermo Fisher, 66837-IG, 1:100), or anti-SC-35 (MilliporeSigma, S4045, 1:1,000) and incubated with the secondary antibodies using Alexa 488-labeled goat anti-mouse IgG (Abcam, Ab150113, 1:1,000), Alexa 594-labeled goat anti-rabbit IgG (Thermo Fisher, A-10012, 1:250). Hoechst 33342 was

used to stain the cell nuclei at 0.2 µg/ml for 15 min. Coverslips were mounted using FlourSave reagent (Milipore, 345789). Images were acquired using a Zeiss confocal microscope LSM900.

## RBP-binding exons

To determine whether sequences surrounding alternative exons are bound by MBNL1, CLIP-seq and RIP-seq data in MBNL Interactome Browser (MIB.amu.edu.pl) [28] were used. MBNL1-binding regions within alternative exons and/or adjacent intron of *Mapt*, *App*, *Ski*, and *Tnik* were investigated. Using the CLIP list [29], alternative exons activated or inhibited by PTBP1 and regulated by FMRP were compared.

## Primers for validation of alternative splicing

| | |
|---|---|
| Dcun1d2 Ex6 RT F | ATGGCTGTTGCATATTGGAAGTT |
| Dcun1d2 Ex6 RT R | CAGGCGGCTTCTAAAGCACT |
| Dcun1d2 Ex2 RT F | GCTCAGAAGGACAAGGTCCG |
| Dcun1d2 Ex2 RT R | TAGACTCTCGGTGAAACGCC |
| Tnik SJC F | CGGCCAGCTGATCTGACG |
| Tnik SJC R | CTCACTGCTCTCCGACTCCT |
| Tnik IJC F | AAGTCCGAAGGATCACCCGT |
| Tnik IJC R | TGCCGTCAGATCCTCATCTAT |
| Tnik Ex20 F | ATCCAGAGACATCACACGGC |
| Tnik Ex22 R | TTCAGGGGGCGGTTTGTTT |
| Tnik Ex25 F | GGCCAAACTCAATGAAGCGA |
| Tnik Ex25 R | GGTGTGTCACTATGAGGGCG |
| Ski IJC F | GTGCCCCGGGTCTCA |
| Ski IJC R | GACGTCTCTTTCTCACTCGC |
| Ski SJC F | CCTGCCACTGGGGCTTC |
| Ski SJC R | AGCCGAGGCTCCGGG |
| Wnk1 SJC F | CAGGGAATACAGCCAACTGTTC |
| Wnk1 SJC R | ACTCCCTGAGTACTCTGTGTTC |
| Wnk1 IJC F | ACCTTGGCTTCATCTGCTACA |
| Wnk1 IJC R | TGAGTACTCTGGTACAAAACATCT |
| App SJC F | TGCTCTGAACAAGCCGAGAC |
| App SJC R | CTGTCGTGGGAAACACGCTG |
| App IJC F | GCAGCGTGTCAACCCAAAG |
| App IJC R | GGGACATTCTCTCTCGGTGC |
| Mapt SJC F | TGAACCAGTATGGCTGACCC |
| Mapt SJC R | GCTGGCCACACGAGCTTTTA |
| Mapt IJC F | TGGCTTAAAAGCCGAAGAAGC |
| Mapt IJC R | TCTTCTCGTCATTTCCTGTCCTG |
| Os9 SJC F | TGGACAAACTCATCAAGAGGCT |
| Os9 SJC R | AATCTTGCCTGTAGGGTGTGG |
| Os9 IJC F | ACCCTACAGAGGAGGAACCTG |
| Os9 IJC R | CAATCTTGCCTTCCGCCGTG |
| Mapt Ex15 F | AAAATCCGGAGAACGAAGCG |
| Mapt Ex15 R | AGGCGGCTCTTACTAGCTGA |
| App Ex2 F | TCGCCATGTTCTGTGGTAAAC |
| App Ex2 R | AATGCAGGTTTTGGTCCCTGA |

*(Continued)*

| | |
|---|---|
| Mbnl1 Ex5 RT F | TGCTCTCGGGAAAAGTGCAA |
| Mbnl1 Ex5 RT R | GGTGGGAGAAATGCTGTATGC |
| Mbnl1 Ex7 RT F | CCAACATGCAGTTACAGCAGC |
| Mbnl1 Ex7 RT R | GTTTTGTGACGACAGCTCTACA |
| Mbnl1 Ex10 F | ATGGTGAGGGAGGGAACTGA |
| Mbnl1 Ex10 R | GGTACTTAAAGCCATGGTGTGC |
| Mbnl1 Ex5 IJC F | AGCTGCCATGACTCAGTCGG |
| Mbnl1 Ex5 IJC R | GAGGAATTCCCAGGTCAAAGGT |
| Mbnl1 Ex5 SJC F | CTACTGCAGCTGCCATGGGAAT |
| Mbnl1 Ex5 SJC R | AAGAGCAGGCCTCTTTGGCAAT |
| Mbnl2 Ex5 RT F | AAACGACAACACCGTAACCG |
| Mbnl2 Ex5 RT F | CTGGCCCCGTTGCTTTTT |
| Mbnl2 Ex9 RT F | CTCAGTTCAACAGCTCCGGT |
| Mbnl2 Ex9 RT R | TGTGACCAGTGGTGTATGGC |
| Slc30a4 Ex2 RT F | AAATGGACCCCTGTGACAACT |
| Slc30a4 Ex2 RT R | GATGGTTCTCTGCACAGCCT |
| Carm1 Ex5 RT F | TACTGCCTACGACCTGAGCA |
| Carm1 Ex5 RT R | CGAGGCCATAGAGATGGCAG |

## Primers for pFlareA reporter cloning

| | |
|---|---|
| NheI-Mapt Ex4-BamHI-F | GCTAGCTAGCTTCTGGGTACA |
| NheI-Mapt Ex4-BamHI-R | CGCGGATCCAAGCGTATCTGTGAC |

## Primers for qPCR

| | |
|---|---|
| Fmr1 F | ACCAAGAAAGAGTCACATTTAACCA |
| Fmr1 R | GGGGTAAAGAAACTTGGGACA |
| Gapdh F | CTACCCCCAATGTGTCCGTC |
| Gapdh R | TGAAGTCGCAGGAGACAACC |
| Mbnl1 F | AGCTGTACTTCCCCCATTGC |
| Mbnl1 R | AGCGGGTGTCATGCACAATA |
| Mbnl2 F | CAACTGTAGACCTGGCCTTCC |
| Mbnl2 R | TCATGGGTACTGTGGGGATGAA |
| Ptbp1 F | AGTGCGCATTACACTGTCCA |
| Ptbp1 R | CTTGAGGTCGTCCTCTGACA |
| Ptbp2 F | TACTAGCTGTTCCAGGGGCT |
| Ptbp2 R | AGGACTGTATTGCCACCAGC |
| hnRNPF F | CCACTCAACCCTGTGAGAGT |
| hnRNPF R | TTGCTAGCCCCTGTTGTTGA |
| hnRNPQ F | AGCCCATGGATACTACTTCAGC |
| hnRNPQ R | ATGTGCAACTAGCCCTGCAA |
| Srsf5 F | GGTGACGATTGAACATGCCC |
| Srsf5 R | CGACTGCTAAAACGGTCGGA |
| Hprt F | GGTTAAGCAGTACAGCCCCA |
| Hprt R | TCCAACACTTCGAGAGGTCC |

### Primers for construction of CRISPR/Cas9 KO cell line

| | | |
|---|---|---|
| mFmr1-sgE3-F | CACCGTATTATAACCTACAGGTGGT | sgRNA for pLenticrispr |
| mFmr1-sgE3-R | AAACACCACCTGTAGGTTATAATAC | sgRNA for pLenticrispr |
| mFmr1 E3 GT F | ACCAAGAAAGAGTCACATTTAACCA | Deletion genotype primer |
| mFmr1 E3 GT R | GGGGTAAAGAAACTTGGGACA | Deletion genotype primer |

## Supporting information

**S1 Fig. Gene Ontology (GO) terms.** (A) GO terms for skipped/included exons in *Fmr1* KO CTX and HC. (B) Volcano plots of RNAs up- or down-regulated ($\log_2$FC > 0.2 or < −0.2, FDR<0.05, $n$ = 3) in *Fmr1* KO liver (LV), muscle (MU), and testis (TE). (C) *Fmr1* RNA levels (TPM) in brain regions and peripheral tissues. (D–F) GO terms of RNAs up- or down-regulated in *Fmr1* KO LV, MU, and TE. (G) GO terms for skipped/included exons in *Fmr1* KO LV and MU.
(TIF)

**S2 Fig.** (A) GO terms for RNAs that display exon skipping in *Fmr1*-depleted N2A cells. (B) DNA sequence analysis of parental N2A cells (top) and Fmr1-depleted cells (bottom). The shaded portion corresponds the amino acid sequences shown in Fig 2. The red box indicates the nucleotides depleted by CRISPR/Cas9 editing. Following editing, the TAG at the right of the shaded box becomes a premature stop codon, leading to nonsense mediated mRNA decay and loss of FMRP expression. (C) Mean fluorescence intensity (MFI) of mCherry/GFP evaluated by flow cytometry in control cells transfected with a splicing reporter with an MBNL1-binding site deletion (UGCUGC). The underlying data can be found in S3 Data.
(TIF)

**S3 Fig. Efficacy of splicing factor depletion and determination of skipping of control exons.** (A–H) Efficacy of *Fmr1* and splicing factor depletion by siRNAs. $^{**}p < 0.01$; $^{***}p < 0.001$; $^{****}p < 0.0001$ ($n$ = 3). (I–T) Determination of skipping of constitutive exons *Mapt* exon 15, *App* exon 2, and *Tnik* exon 25 following *Fmr1* and splicing factor depletion. $^{*}p < 0.05$ ($n$ = 3). The underlying data can be found in S3 Data.
(TIF)

**S4 Fig. MBNL1-binding sites in exons regulated by FMRP.** Analysis of MBNL1-binding sites was conducted using data from MBNL1 RIP-seq, with binding sites highlighted in blue, and CLIP-seq, with binding sites indicated in orange [28,29]. These binding sites were analyzed in exons located near *Fmr1*-regulated splicing events that display alterations which are shaded in gray.
(TIF)

**S5 Fig. PTBP1-binding sites in exons regulated by FMRP.** Analysis of PTBP1-binding sites using 6-mer binding motifs TCTCTC and CTCTCT. The binding sites are highlighted in red and green, respectively. These binding sites were analyzed in exons located near *Fmr1*-regulated splicing events that display alterations which are shaded in gray.
(TIFF)

**S6 Fig. Western blot for SETD2 and RT-PCR for *Mbnl2*.** (A) Western blot for SETD2, H3K36me3, H3, and FMRP in nuclear extracts of control and *Fmr1*-deficient N2A cells. LaminB1 is used as a loading control. (B) RT-PCR for *Mbnl2* exon 5 splicing in control and

*Fmr1*-deficient N2A cells and WT and *Fmr1* KO HC. Band intensity of exon 5 is quantified and mean ± SD is shown (Student's *t* test, *$p < 0.05$). The constitutive exon 9 was amplified to compare total *Mbnl2* RNA expression. The underlying data can be found in S3 Data.
(TIF)

**S7 Fig. Efficacy of FMRP and MBNL1 depletion and ectopic expression.** Western blot for FMRP and MBNL1. GAPDH is used as a loading control.
(TIF)

**S8 Fig. Splicing events mediated by MBNL1 isoforms.** (A) Violin plots showing the distribution of delta PSI in siFmr1+nMBNL1 and siMbnl2+cMBNL1. The solid line is the median and the dashed lines are quartiles. *P*-value < 0.05, |delta PSI| > 0.05. (B) PSI of *Mbnl1* exon 5 in all experimental groups. (C) Venn diagram comparing skipped exon events in siFmr1 and siMbnl1 cells. The underlying data can be found in S3 Data.
(TIF)

**S9 Fig. FMRP-mediated regulation of RNA expression and splicing.** (A) Heatmap visualizing the PSI values of SEs in all groups that, when compared to controls, pass a threshold for significant change upon *Fmr1* depletion (*P*-value < 0.05, |delta PSI| > 0.05). PSI values were adjusted by z-score. (B) RT-PCR for *Slc30a4* exon 2 and *Carm1* exon 5 splicing. (C) Volcano plots showing the differential expression of RNAs associated with RNA splicing upon *Fmr1* depletion. $Log_2FC > 0.2$ or $< -0.2$, FDR < 0.05, *n* = 3. (D) Volcano plots showing the AS events involved in RNA splicing upon *Fmr1* depletion. *P*-value < 0.05, |delta PSI| > 0.05.
(TIF)

**S1 Data. MBNL1-binding sites analysis on FMRP target RNAs.**
(XLSX)

**S2 Data. MBNL1-binding sites analysis on FMRP non-target RNAs.**
(XLSX)

**S3 Data. Excel spread sheet containing numerical values used in figures (Figs 1F, 1G, 2C, 2F–2H, 3B–3D, 4A–4C, 5B–5D, 7B–7E, 8B–8C, 8H, 9B, 9D, 9G, 9H, S2C, S3A–S3T, S6B, S8A and S8B).** The processed RNA-seq data supporting the figures is available in the GEO database (Figs 1B, 1D, 1I, 8E, 8F, S1A, S1B, S1C–S1G, S2A, S9A and S9B).
(XLSX)

**S1 Raw Images. Original images for Figs 1G, 2A, 2E, 2H, 4B, 4C, 5B, 5C, 7B, 7C, S6A, S6B, S7A and S9B.**
(PDF)

## Acknowledgments

We thank Dr. Guey Shin Wang (National Yang-Ming University and Academia Sinica, Taipei, Taiwan) for sharing FLAG-MBNL1$^{Ex5}$, FLAG-MBNL1$^{\Delta Ex5}$, and control plasmid. We thank Tammy Krumpoch and Susanne Pechhold (UMass Chan Medical School Flow cytometry core) for helpful discussions. We thank Dr. Ozkan Aydemir and Ms. SitharaRaju Ponny for advice on bioinformatic and statistical analysis and Dr. Heleen van't Spijker for assistance with immunofluorescence microscopy. We thank Dr. Mariya Ivshina for insightful conversations related to this study and Drs. Pablo Visconti and Maria Gracia Gervasi (University of Massachusetts Amherst) for assistance with some initial experiments.

## Author Contributions

**Conceptualization:** Suna Jung, Sneha Shah, Joel D. Richter.

**Data curation:** Suna Jung, Geongoo Han.

**Formal analysis:** Suna Jung, Sneha Shah, Geongoo Han.

**Funding acquisition:** Sneha Shah, Joel D. Richter.

**Investigation:** Suna Jung, Sneha Shah.

**Methodology:** Suna Jung.

**Project administration:** Suna Jung, Sneha Shah.

**Supervision:** Joel D. Richter.

**Validation:** Suna Jung.

**Visualization:** Suna Jung.

**Writing – original draft:** Suna Jung, Sneha Shah, Joel D. Richter.

**Writing – review & editing:** Suna Jung, Sneha Shah, Joel D. Richter.

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
