## [Editor Report · Decision Letter 0]

7 Jan 2023

Dear Dr Richter, 

Thank you for submitting your manuscript entitled "FMRP-Regulated RNA Splicing Involves Multiple RNA Binding Proteins and Translational Control of Mbnl1 RNA" for consideration as a Research Article by PLOS Biology.

Your manuscript has now been evaluated by the PLOS Biology editorial staff, as well as by an academic editor with relevant expertise, and I am writing to let you know that we would like to send your submission out for external peer review.

Once your full submission is complete, your paper will undergo a series of checks in preparation for peer review. After your manuscript has passed the checks it will be sent out for review. To provide the metadata for your submission, please Login to Editorial Manager (https://www.editorialmanager.com/pbiology) within two working days, i.e. by Jan 09 2023 11:59PM.

Kind regards,

Richard

Richard Hodge, PhD

Associate Editor, PLOS Biology

rhodge@plos.org

PLOS

---

## [Decision Letter · Decision Letter 1]

31 Jan 2023

Dear Dr Richter,

Thank you for your patience while your manuscript "FMRP-Regulated RNA Splicing Involves Multiple RNA Binding Proteins and Translational Control of Mbnl1 RNA" was peer-reviewed at PLOS Biology. Your manuscript has been evaluated by the PLOS Biology editors, an Academic Editor with relevant expertise, and by three independent reviewers.

The reviews are attached below. As you will see, the reviewers think that your findings are potentially interesting and important, but they raise overlapping concerns with the overall strength of the evidence to support a direct role for Mbnl1 in regulating the downstream splicing events. Specifically, Reviewer #1 notes that mutational analysis of the relevant binding motifs should be included and Reviewer #2 asks for RNA-sequencing data to provide more direct support for the claim that a loss of nuclear Mbnl1 mediates the splicing changes. 

After discussions with the Academic Editor, it is clear that a substantial amount of work would be required to meet the criteria for publication in PLOS Biology. However, given our and the reviewer interest in your study, we would be open to inviting a comprehensive revision of the study that thoroughly addresses all the reviewers' comments, including the mutational analysis and RNA-seq data. Given the extent of revision that would be needed, we cannot make a decision about publication until we have seen the revised manuscript and evaluated your response to the reviewers' comments. Your revised manuscript would need to be seen by the reviewers again, but please note that we would not engage them unless their main concerns have been addressed.

We appreciate that these requests represent a great deal of extra work, and we are willing to relax our standard revision time to allow you 6 months to revise your study. Please email us (plosbiology@plos.org) if you have any questions or concerns, or envision needing a (short) extension.

**IMPORTANT - SUBMITTING YOUR REVISION**

*Resubmission Checklist*

*Published Peer Review*

*PLOS Data Policy*

*Blot and Gel Data Policy*

Sincerely,

Richard

Richard Hodge, PhD

Associate Editor, PLOS Biology

rhodge@plos.org

REVIEWS:

Reviewer #1: In this manuscript, Jung and colleagues interrogated RNA dysregulation in FMRP-deficient mice and cultured cells, focusing on differences in alternative splicing regulation. They found that in FMRP-deficient cells, the inclusion rates of exons within transcripts encoding the splicing factors MBNL1 and PTBP1 was altered. Using this data, they postulate that this alters the activities of MBNL1 and PTBP1, leading to further downstream changes in the splicing of other transcripts. Although the data presented is interesting and suggestive, it is not enough to demonstrate that any of the changes in splicing observed are directly due to perturbations of MBNL1 or PTBP1. Further experiments are needed to justify this claim.

MAJOR COMMENTS

1. My main concern comes from the authors' repeated ascribing of splicing regulatory effects by specific RBPs solely because (i) the inclusion of an exon changed in response to perturbation of the RBP and (ii) the exon is flanked by motifs that match the RBP's consensus binding site. As examples, this logic is laid out in figure 3A and in the text describing figure 4A. Showing that an RBP regulates the splicing of an exon requires both that the exon be differentially included upon RBP perturbation and that the effect requires binding of the RBP to the regulated transcript. Otherwise, the effect could be due to indirect effects that do not at all involve the RBP of interest.

Using motif presence alone as evidence for regulation by an RBP is problematic for a few reasons. First, for any given RBP (and MBNL1 specifically, e.g. PMID 27720642), the vast majority of their consensus motifs are not actually bound by the protein in cells. This can be seen by comparing CLIP binding sites across the transcriptome to motif occurrences. The former is a small subset (often only 10-20%) of the latter. 

Second, most RBP motifs are quite small. As an example, the consensus motif for MBNL1 is UGCU. As a 4mer, this motif would be expected to occur by random chance once every 256 nucleotides (using a simplistic model). In the examples presented in figure 3A, it looks like the abundances of motifs near the regulated exons may not be too different from the null expectation.

There are a few ways the authors could bolster their claims. First, they could use published CLIP datasets for MBNL1 (e.g. PMID 27733504) and/or PTBP1 (e.g. PMID 27926877) to ask if sequences surrounding their favorite exons are actually bound by these RBPs. Expanding this, they could ask transcriptome-wide if exons that are differentially regulated upon FMRP loss are more likely to have MBNL1/PTBP1/etc CLIP peaks near them than FMRP-insensitive exons.

Similarly, if the authors wish to focus on RBP motifs rather than CLIP evidence, they could show that these motifs are actually enriched near regulated exons that simply present near them. As an example of this, when describing Data S9, the authors state that "50% of skipped or included exons in N2A cells contain binding sites for MBNL1 using RBPmap." This is fine, but how does that compare to a set of control alternative exons whose inclusion was unchanged? As stated above, given the high likelihood of these motifs to appear by chance, this comparison is a key control.

Finally, an alternative way to more directly show that these RBPs are regulating specific exons is to mutate the motifs they identified (e.g. those in Figure 3A) and ask if that alters the inclusion of the exon of interest, ideally in the same direction as perturbation of the cognate RBP. This could potentially be done using the splicing reporter described in Figure 2G.

2. In figure 6, the authors analyze the inclusion of alternative exons with Mbnl1 RNA in response to FMRP perturbation from a range of different samples, arguing that because specific exons (e.g. exon 5 and exon 7) are repeatedly identified as FMRP-sensitive across a range of samples, the effect is reproducible. However, the direction of the regulation for a given exon is different between many of the samples (look at Inclusion difference [a.k.a. delta PSI] column). To me, this argues that the regulation of Mbnl1 splicing is not directly due to FMRP and that these highly regulated exons themselves are being indirectly regulated by something else.

3. All of the RT-PCR experiments to quantify alternative splicing are nice to see, but it could be helpful to show metrics for those exons from the RNAseq data (e.g. delta PSI, FDR, etc.) Seeing that the RNAseq/rMATS analysis and RT-PCR generally agree would lend confidence to the results.

4. In multiple places, when the authors are describing cutoffs for identifying significant exons to focus their analysis on, the metrics presented are confusing (e.g. log2FC p < 0.05, PSI > 0.05). The software the authors used, rMATS, does not report log2FC. log2FC is very rarely used in splicing analysis. A much better metric would be "delta PSI" which compares the average PSI value for a given exon between the two tested conditions. In rMATS output, this is the IncLevelDifference column. rMATS then computes a multiple-hypothesis corrected FDR value for if the two sets of PSI values for an exon are different across conditions. This may be what the authors mean rather than log2FC p. 

Similarly, it does not make sense to use "PSI" as a metric here since that is the value for an exon in one condition, when what you want here are exons that are differentially included between two conditions. Thus the term "delta PSI", which is the difference in PSI values between conditions, is normally used. 

Finally, in order to study both exons that are more included and more skipped, it is common to set the delta PSI cutoff at abs(0.05), i.e. > 0.05 and < -0.05.

All of this may be what the authors actually did, but the way it is communicated is confusing and likely incorrect.

5. Similar to point 4, the utility of figure 1F is unclear. It would be more useful to plot delta PSI between the wildtype and KO conditions. I know that PSI, and not delta PSI is currently being plotted because PSI ranges between 0 and 1 and delta PSI ranges between -1 and 1. Plotting delta PSI would allow the visualization of broad, coordinated changes in alternative isoforms, for example a trend to global inclusion or exclusion of alternative exons.

6. Is there an overall correlation in gene expression and splicing changes between the N2A Fmr1 knockdown and knockout experiments? This would lend further confidence to this data.

MINOR COMMENTS

1. In multiple places in the text, the authors use the term "Mbnl1 RNA self-splicing" or variants thereof to describe the known regulatory effect that MBNL1 protein has on the splicing of the RNA that encodes it. I understand what the authors are trying to say, but the term "RNA self-splicing" already has a very specific meaning (e.g. group II intron self-splicing) that is quite different from what is being discussed here. Rephrasing this could improve the clarity of the manuscript.

2. In the introduction, the sentence "Splicing factors MBNL1, PTBP1, and HNRNPF are responsible for altered splicing in Fmr1-deficient cells. FMRP regulates the translation of two of these factors, MBNL1 and HNRNPQ." This is confusing as HNRNPQ is not one of "these" factors mentioned in the previous sentence.

3. When describing figure 3B, the authors state that "Because the magnitude of Mapt exon 4 inclusion was additive when both Mbnl1 and Fmr1 were depleted, we surmise that a second splicing factor under the control of FMRP is involved in this splicing event."

This may be true, but it could also be true that residual Mbnl1 RNA that survives the Mbnl1 knockdown is then further misregulated upon Fmr1 knockdown (perhaps translationally, as the authors show). This would give an additive effect without the need to invoke a second RBP.

Reviewer #2: The manuscript interrogates the factors that might mediate the changes in RNA Splicing that are observed upon deficient FMRP activity. The authors previously reported a correlation between the up-regulation of SETD2, altered H3K36me3 chromatin marks, and RNA splicing mis-regulation in Fmr1-deficient mouse brain. Now they interrogated splicing changes in more depth with Fmr1 KO N2A cells, and detected no alteration in SETD2 levels, concluding that to splicing dysregulation is more likely driven by factors other than a change in H3K36me3. The study focuses on the splicing factors hnRNPF, PTPB1 and MBNL1, which are candidate regulators of specific observed splicing changes. The study focuses mostly on MBNL1, because FMRP directly regulates the translation of Mbnl1 mRNA, which can explain a subset of downstream splicing changes including altered Mbnl1 self-splicing, and altered Mbnl1 self-splicing also occurs in human FXS post-mortem brain. This is a potentially valuable study showing how regulation of translation and RNA processing might be coupled via cross-regulation between RNA-binding proteins. However, the data supporting the conclusions is quite limited at present, and more thorough investigation with RNAseq would be needed.

Major suggestions

The study finds a strong increase in the abundance of MBNL1 and hnRNPQ upon Fmr1 depletion, but MBNL1 shows a slightly decreased tendency to localise to the nucleus, possibly due to its splicing shift. For now, I have a hard time seeing much change in nuclear abundance of MBNL1, or estimating how Fmr1 should be expected to affect its splicing activity. It seems that the slight change in nuclear transport is compensated by increased abundance of MBNL1. Iit would be helpful if absolute abundance of MBNL1 in the nucleus could be quantified and presented.

Lack of clear effect on nuclear MBNL1 levels makes it difficult to understand how MBNL1 might mediate splicing changes caused by Fmr1 depletion. Moreover, evidence from a few chosen exons is also limited -, then co-depletion of Mbnl1 and Fmr1 only shows an additive effect for Mapt Exon 4. In addition to co-depletion, authors should also perform experiments where Fmr1 depletion is coupled with transfection of cDNA expressing one or the other MBNL1 isoform, to show that only transfection of the NLS-containing isoform can rescue Mapt Exon 4 splicing back to control levels.

If the above is the case, the authors should perform RNAseq analysis of cells treated with various siRNA or co-transfection combinations to show more thorough evidence for the claim that loss of nuclear MBNL1 mediates a subset of splicing changes induced by Fmr1 depletion.

Minor suggestions

I find the statement 'FMRP-Regulated RNA Splicing' in the title potentially misleading, because such statements normally refer to direct regulation. FMRP doesn't regulate RNA Splicing - instead it regulates translation. I'd find a title along these lines clearer: "FMRP Affects Splicing by Regulating Translation of Multiple Splicing Factors, Including Mbnl1"

The abstract states: When FMRP is lost, SETD2 levels are excessive, which alters the H3K36me3 landscape and secondarily, alternative pre- mRNA splicing. - however, the paper rejects this statement with the conclusion "a changed chromatin landscape and altered splicing in FMRP- deficient cells may not be linked", but one wouldn't realise this by reading the abstract. I'd find it more informative if SETD2 was mentioned at results, rather than introduction of the abstract, but stating sth along: We find that the previously reported changes in chromatin landscape may not be linked to altered splicing in FMRP-deficient cells, but instead specific RNA binding proteins…"

The abstract also states: "Here we show that in Fmr1-deficient mice, RNA mis-splicing occurs in several brain regions and peripheral tissues." This makes an impression that we're looking at a new finding, but a similar observation was already made previously by the authors (https://pubmed.ncbi.nlm.nih.gov/32234480/). The abstract should make it clearer what was known and what is new in terms of discovered splicing changes.

Reviewer #3: In this manuscript, Jung and colleagues study the role of FMRP, a protein that is absent in Fragile X Syndrome (FXS), in pre-mRNA splicing.

First, the authors analysed gene expression in an Fmr1 KO mouse model, focusing on brain tissues (hippocampus, cerebellum, cortex), as well as peripheral tissues (liver, muscle, and testes). This analysis revealed extensive mis-regulation of gene expression and importantly, of Alternative splicing (AS), predominantly affecting exon skipping (SE). These data are of interest, and expands vastly previous observations by the authors, published in their Cell rep paper in 2020.

This analysis was complemented by the use of mouse N2a neuroblastoma cells, that were depleted of Fmr1 using siRNA. As seen in mouse tissues, the authors again observed a role for FMRP in the regulation of AS, predominantly affecting exon inclusion-skipping mode of AS. To validate their results, the authors decreased FMRP expression using genome editing of the Fmr1 pre-mRNA, resulting an in-frame stop codon leading to NMD. Importantly, this strategy resulted in complete loss of FMRP protein. Interestingly, they observed almost identical changes in AS, as those observed following. siRNA-mediated depletion for the Mapt4 mRNA. Next, they developed a Mapt4-based splicing reporter.

Next, Jung and co-authors focused on splicing factors (SFs) that regulate AS in response to FMRP levels. For this, they use the SFMap database to identify putative SFs binding sites. This predicted binding sites for MBNL1, PTBP1, hnRNP F and hnRNP Q in the vicinity of Mapt exon 4, App exon 8 and Tnik exon 21, all of which are regulated by FMRP. They show that FMRP binds to Mbnl1 and also Ptbp1 mRNA. In a series of interesting experiments, the authors determine that FRMP has a dual role in regulating the expression of MBNL1, by affecting splicing of its pre-mRNA, as well as its mRNA translation. In particular, FMR1 promotes exon 5 inclusion in Mbnl1, which encodes an NLS. Thus, in the absence of FMR1, E5 skipping will lead to a predominantly cytoplasmic MBNL1, affecting regulation of splicing.

In summary, there is a lot of work here, the experiments have been carefully designed and most of the conclusions are well supported by the data. In summary, this is an interesting paper, which highlights that FMRP-controlled mRNA translation and splicing leads to extensive dysregulation of gene expression in the absence of FMRP and could explain phenotypes observed in FXS.

Specific comments

1- The following statement in the Abstract is extremely confusing, mostly taken into account that there are no experiments on Setd2 

"In the brain, FMRP stalls ribosomes on specific mRNAs including Setd2, whose encoded protein catalyzes the epigenetic mark H3K36me3. When FMRP is lost, SETD2 levels are excessive, which alters the H3K36me3 landscape and secondarily, alternative premRNA splicing" 

This should be moved to the Introduction and the authors should explain better the discrepancy with their own previous results showing an indirect effect of FMRP in splicing via SETD2/H3K36me3

2- While interesting, results presented in Fig 3 show that FMRP works together with a limited subset of SFs, but this subset was defined by a predictive tool. It still remains possible, that FMRP may work together with other SFs that have not been tested here. This should be discussed.

3- Have the experiments following genome editing in Figs 4E, H been extended to the analysis of other mRNAs, besides Mapt4?

---

## [Decision Letter · Decision Letter 2]

26 Sep 2023

Dear Dr Richter,

Thank you for your patience while we considered your revised manuscript "FMRP-Regulated Alternative Splicing is Multifactorial and Resembles Splicing Control by MBNL1" for consideration as a Research Article at PLOS Biology. Your revised study has now been evaluated by the PLOS Biology editors, the Academic Editor and the original reviewers.

In light of the reviews, which you will find at the end of this email, we are pleased to offer you the opportunity to address the remaining points from Reviewer #2 in a revision that we anticipate should not take you very long. This includes repeating the experiment presented in Figure 5C to ensure equal loading between the conditions, reporting a single heatmap in Figure 9E and performing qPCR validation of the RNA-seq data in Figure 9G. We will then assess your revised manuscript and your response to the reviewers' comments with our Academic Editor aiming to avoid further rounds of peer-review, although might need to consult with the reviewers, depending on the nature of the revisions.

In addition, I would be grateful if you could please address the following editorial and data-related requests that I have provided below (A-I):

(A) We would like to suggest the following modification to the title: 

“FMRP deficiency leads to extensive dysregulation of splicing due to exon skipping of MBLN1 that mislocalizes it to the cytoplasm”

(B) In the animal ethics statement in the Methods section, please include the full name of the IACUC/ethics committee that reviewed and approved the animal care and use protocol/permit/project license. Please also include an approval number.

(C) You may be aware of the PLOS Data Policy, which requires that all data be made available without restriction: http://journals.plos.org/plosbiology/s/data-availability. For more information, please also see this editorial: http://dx.doi.org/10.1371/journal.pbio.1001797

-Supplementary files (e.g., excel). Please ensure that all data files are uploaded as 'Supporting Information' and are invariably referred to (in the manuscript, figure legends, and the Description field when uploading your files) using the following format verbatim: S1 Data, S2 Data, etc. Multiple panels of a single or even several figures can be included as multiple sheets in one excel file that is saved using exactly the following convention: S1_Data.xlsx (using an underscore).

-Deposition in a publicly available repository. Please also provide the accession code or a reviewer link so that we may view your data before publication. 

Figure 1B, 1D, 1F, 1G, 1I, 2C, 2F-H, 3B-D, 4A-C, 5B-D, 7B-E, 8B-F, 8H, 9B, 9D, 9G-H, S1A-G, S2A, S2C, S3A-T, S6B, S8A-B, S9A-B

(D) Thank you for depositing the RNA-seq data in the GEO database (GSE207145). However, I note that the data is currently on hold until Jan 1st 2024. We ask that you please make this data publicly available before publication.

(E) Please also ensure that each of the relevant figure legends in your manuscript include information on *WHERE THE UNDERLYING DATA CAN BE FOUND*, and ensure your supplemental data file/s has a legend.

(F) Thank you for already providing the original and uncropped images supporting the blot and gel results reported in the Figures. However, we note that S4A-B may be mislabelled in the ‘Original Images’ file (should be S6A-B?)

(G) Please ensure that your Data Statement in the submission system accurately describes where your data can be found and is in final format, as it will be published as written there. 

(H) Please note that per journal policy, the model system/species studied should be clearly stated in the abstract of your manuscript. 

(I) Please also provide a blurb which (if accepted) will be included in our weekly and monthly Electronic Table of Contents, sent out to readers of PLOS Biology, and may be used to promote your article in social media. The blurb should be about 30-40 words long and is subject to editorial changes. It should, without exaggeration, entice people to read your manuscript. It should not be redundant with the title and should not contain acronyms or abbreviations. For examples, view our author guidelines: https://journals.plos.org/plosbiology/s/revising-your-manuscript#loc-blurb

We expect to receive your revised manuscript within 2 months. Please email us (plosbiology@plos.org) if you have any questions or concerns, or would like to request an extension. 

**IMPORTANT - SUBMITTING YOUR REVISION**

*Resubmission Checklist*

*Published Peer Review*

*PLOS Data Policy*

*Blot and Gel Data Policy*

Sincerely,

Richard

Richard Hodge, PhD

rhodge@plos.org

REVIEWS:

Reviewer #1: The authors have satisfactorily responded to my requests.

Reviewer #2 (Jernej Ule, signs review): The authors have performed all key requested experiments, especially RNAseq in combination with rescue using the nuclear or cytoplasmic variants of MBNL1. Fig. 9D shows high correlation between knockdowns of Fmr1 and Mbnl1, which is in support of the model. I congratulate the authors on this complex and interesting work, and my remaining requests are relatively minor and hopefully achievable:

1. In Figure 5C, the western blot of nuclear MBNL1 is normalised to nuclear Lamin B. However, the amount of LaminB1 signal is dramatically higher in the siFmr1 condition, and it seems that the signal might be saturated. It would be interesting to know whether the higher LaminB1 signal is due to inappropriate quantification of protein (and thus unequal loading), or does it reflect a change in LaminB1 abundance in siFmr1 condition? Either way, this experiment should be redone by ensuring that the loading marker shows similar signal in both conditions - if needed, another nuclear marker (histone?) could be used that doesn't change. For now, the nuclear MBNL1 signal seems increased by eye pre-normalisaton, and it remains possible that there is a normalisation artefacts caused by LaminB1 signal saturation in siFmr1 condition. Since the drop in nuclear Mbnl1 abundance is a key aspect of the proposed model, I believe the authors should make some more effort to present convincing evidence.

2. Fig 9E shows that cotransfection of nMbnl1 can rescue many splicing defects of siFmr1, and siMbnl1 can recapitulate many of the changes. However, only those events are shown where rescue/recapitulation is observed, so the heatmap can't inform on the extent of the relationship. Instead of showing two separate heatmaps in Fig 9E/F, a single heatmap should be shown that includes all SEs that pass a certain threshold for significant change upon siFmr1, and all conditions are shown in separate columns (including siMbnl1). 

3. Fig 9g shows PSI values of representative spliced exons, and as I understand this is showing RNAseq data. Since these RNAseq results provide key evidence for the model, it would be helpful to validate a few exons with a different method. In particular, few exons from current Fig 9E could be randomly selected (i.e., those where Mbnl1 expression seems to rescue the change upon siFmr1) and validated with RT-PCR & electrophoresis (including siFmr1, siMbnl1, and rescue of both with nuclear Mbnl1 if possible). 

Reviewer #3: The authors performed a thorough revision and have addressed the minor concerns and suggestions that I raised during the first round of review. Thus, I remain enthusiastic about this study. I do recommend publication in PLoS Biol.

---

## [Editor Report · Decision Letter 3]

3 Nov 2023

Dear Joel,

Thank you for the submission of your revised Research Article "FMRP deficiency leads to multifactorial dysregulation of splicing and mislocalization of MBNL1 to the cytoplasm" for publication in PLOS Biology. On behalf of my colleagues and the Academic Editor, Tom Misteli, I am pleased to say that we can accept your manuscript for publication, provided you address any remaining formatting and reporting issues. These will be detailed in an email you should receive within 2-3 business days from our colleagues in the journal operations team; no action is required from you until then. 

Please note that we will not be able to formally accept your manuscript and schedule it for publication until you have completed any requested changes. This would include making the RNA-seq data deposited in the GEO (GSE207145) publicly available, as I note it is still on hold and scheduled for release on Jan 1st 2024. I would be grateful if you could please release this data during the production process. 

PRESS

Best wishes, 

Richard

Richard Hodge, PhD

rhodge@plos.org

PLOS
